# LMC: Large Model Collaboration with Cross-assessment for Training-Free Open-Set Object Recognition

**Haoxuan Qu** [*]
SUTD
Singapore
haoxuan_qu@mymail.sutd.edu.sg

**Xiaofei Hui** [*]
SUTD
Singapore
xiaofei_hui@sutd.edu.sg

**Yujun Cai**
Meta
U.S.
yujuncai@meta.com

**Jun Liu** [†]
SUTD
Singapore
jun_liu@sutd.edu.sg

## Abstract

Open-set object recognition aims to identify if an object is from a class that has been encountered during training or not. To perform open-set object recognition accurately, a key challenge is how to reduce the reliance on *spurious-discriminative* features. In this paper, motivated by that different large models pre-trained through different paradigms can possess very rich while distinct implicit knowledge, we propose a novel framework named **L**arge **M**odel **C**ollaboration (**LMC**) to tackle the above challenge via collaborating different off-the-shelf large models in a training-free manner. Moreover, we also incorporate the proposed framework with several novel designs to effectively extract implicit knowledge from large models. Extensive experiments demonstrate the efficacy of our proposed framework. Code is available here.

## 1 Introduction

Open-set object recognition aims to identify if an object is from a *closed-set class* that has appeared during training, or an *open-set class* that has not been encountered in the training set. It is a crucial task since *open-set classes* always exist in the real world [63], and misidentifying *open-set classes* into *closed-set ones* can lead to severe consequences in many practical applications [2], such as autonomous driving [19], robotics system [54], malware classification [18], and medical analysis [15]. Due to its importance, open-set object recognition has received a lot of research attention recently and various methods have been proposed [14, 36, 28, 6, 66, 56, 12].

To perform open-set object recognition well, some existing works [34, 40] have noted that a key challenge is how to reduce the reliance on *spurious-discriminative* features. Specifically, a *spurious-discriminative* feature refers to a feature that satisfies the following two requirements: 1) it is discriminative between a closed-set class and other closed-set classes; 2) it is shared between this closed-set class and certain open-set classes. As suggested by previous works [34, 40], while such *spurious-discriminative* features can facilitate classification within the closed set, in open-set recognition, relying on such features can make closed-set and open-set classes to be easily confused

---

[*]Both authors contributed equally to the work.
[†]Corresponding Author

37th Conference on Neural Information Processing Systems (NeurIPS 2023).

with each other, and thus bring negative effects in accurately identifying open-set objects. For example, as shown in Fig. 1, given a closed set of classes including bighorn, tabby cat, and hog, within these closed-set classes, "with horn" as a feature of bighorn can be used to discriminate bighorn from other classes (tabby cat and hog). However, relying on "with horn" to perform open-set object recognition, bighorn can be easily confused with potential open-set classes such as ox and gazelle that are also "with horn". Therefore, it can be difficult to correctly identify ox and gazelle as open-set objects. To avoid open-set objects from being misidentified into closed-set classes, a common type of methods is to simulate additional virtual open-set samples or classes [14, 36, 67, 12, 34]. Specifically, among this type of methods, various external knowledge bases have been used, such as pre-trained GloVe embeddings [67] and extra datasets [12]. Yet, despite the considerable efforts made and various architectures specifically designed for this task, the open-set object recognition task remains challenging. This is because, in the real world, various different sorts of open-set classes can appear [11]. Moreover, given a trained classifier, to further equip it with the ability to identify open-set objects, most existing open-set recognition methods require an extra training process. Especially when the classifier has been deployed, conducting such an extra training process can be inconvenient, as it needs both additional training time and access to training data. Note that the training data can be no longer accessible during deployment (e.g., due to privacy concerns [35, 22]).

Recently, large models containing very rich implicit knowledge [17] have served as powerful external knowledge bases in many areas [13]. For example, ChatGPT [38] possesses rich common-sense knowledge, and has been applied in various natural language processing tasks [42, 20]. DALL-E [47], which holds strong image-generative ability, has been used in different generation tasks [32, 26]. CLIP [45] has been leveraged for visual recognition tasks [72, 41] since it can align relevant image-text pairs. Besides, being capable of aligning relevant image patches, DINO [5] has been applied in areas [55, 1] such as image retrieval.

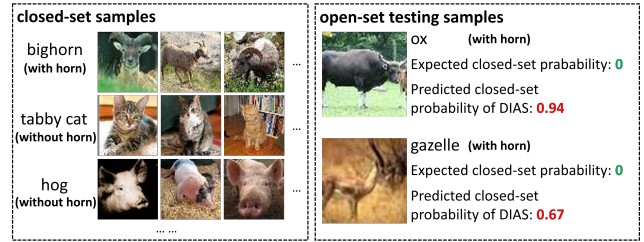

Figure 1: Illustration of a *spurious-discriminative* feature example. As shown, given "with horn" a *spurious-discriminative* feature of the closed set, DIAS [34] as a recently proposed open-set recognition method misidentifies ox and gazelle (both with horn) as closed-set classes with high probability.

Inspired by that large models often contain rich knowledge, here we aim to investigate *leveraging such rich knowledge to handle the open-set object recognition task.* However, using large models to perform open-set object recognition is non-trivial due to the following challenges: 1) It can be difficult to directly leverage an existing large model to effectively perform open-set object recognition. Take CLIP as an example, being able to align an image with a closed set of classes and thus finding out the most relevant class in the set, CLIP has been applied in various visual recognition tasks, such as zero-shot and few-shot image classification [45]. However, CLIP cannot handle the open-set object recognition task effectively by itself. This is because, various sorts of *open-set classes* exist in the real world, and it is difficult to know which class should the testing image be aligned with when using CLIP. 2) Meanwhile, as the knowledge in large models is often implicit [57, 3], it can be challenging to extract the desired knowledge (e.g., in our case, the knowledge that can facilitate open-set object recognition) from large models successfully [57]. To handle these challenges, in this work, with the observation that different large models pre-trained through different paradigms can contain distinct knowledge, we first propose a novel open-set object recognition framework named **L**arge **M**odel **C**ollaboration (**LMC**). As shown in Fig. 2, the proposed LMC performs open-set object recognition in a training-free manner by leveraging rich and distinct knowledge from different off-the-shelf large models complementarily. Moreover, LMC is also incorporated with several novel designs to facilitate extracting implicit knowledge in large models effectively.

Overall, to better perform open-set object recognition, LMC aims to reduce the reliance of this task on *spurious-discriminative* features. To achieve this, LMC proposes to simulate additional virtual open-set classes that share the *spurious-discriminative* features through the collaboration of different large models. Specifically, in LMC, motivated by that ChatGPT possesses rich common-sense knowledge, we first adopt ChatGPT to expand the list of closed-set classes with additional virtual

open-set classes. During this process, to extract implicit knowledge from ChatGPT effectively, we propose to guide ChatGPT with intermediate reasoning, while also enabling ChatGPT to perform self-checking on whether it covers as many *spurious-discriminative* features as possible. Moreover, inspired by that images can catch different details that are complementary to the text domain [60], in addition to representing classes in the expanded list with their names in the text domain, we propose to further use image information to better represent these classes. Specifically, to facilitate extracting image information from large models, we propose to incorporate our framework with a novel cyclic module, which collaborates ChatGPT, DALL-E, and CLIP through a cross-assessing mechanism to generate diverse images for each class. Once both the text and image information have been extracted for each class in the expanded list, during inference, we propose to use the image-text alignment ability of CLIP and the image-image alignment ability of DINO in collaboration to perform open-set object recognition on testing images.

The contributions of our work are summarized as follows. 1) We propose LMC, a novel framework that can handle the open-set object recognition task in a training-free manner, via collaborating different off-the-shelf pre-trained large models and leveraging their knowledge in a complementary manner. 2) We introduce several designs in LMC to extract both text and image information from large models effectively. 3) On the evaluated benchmarks, LMC achieves state-of-the-art performance.

## 2   Related Work

**Open-set Object Recognition.** Due to the wide range of applications, open-set object recognition has received a lot of research attention [48, 4, 50, 14, 63, 39, 68, 53, 51, 7, 6, 33, 23, 56, 61, 19, 58, 70, 31, 71, 49, 21]. Bendale and Boult [4] made the first attempt of applying deep neural networks into the task of open-set recognition. Specifically, they first showed that simply thresholding on the softmax probability cannot yield a robust open-set recognition, and thus proposed Openmax based on the Extreme Value Theory. Ge et al. [14] further proposed G-Openmax to generate unknown samples using generative models. After that, Neal et al. [36] showed the effectiveness of generating images that do not belong to any of the closed-set classes but look similar to those images in the training dataset. Later on, OpenGAN, which uses real open-set images to perform model selection, was proposed by Kong et al. [23]. Vaze et al. [56] proposed to well-train the closed-set classifier so that it can perform both closed-set image classification and open-set recognition well. Besides, Esmaeilpour et al. [12] proposed to train an image description generator using an extra image captioning dataset and use the trained generator to improve the open-set object recognition performance.

Different from existing approaches, in this work, from a novel perspective, we investigate how to collaborate different large models to perform open-set object recognition in a training-free manner. Besides, we also introduce several designs to extract implicit knowledge from large models effectively.

**Large Models.** Recently, a variety of large models have been proposed, such as ChatGPT [38], DALL-E [47], CLIP [45], and DINO [5]. Such large models contain rich knowledge and have been studied in various tasks [20, 65, 69, 32, 64, 27, 1], such as language translation [20], video generation [32], and image retrieval [1]. In this work, we design a new framework to handle the open-set object recognition task by cross-assessing among large models and leveraging the rich and distinct knowledge from different large models in a complementary manner.

## 3   Proposed Method

Given a testing image from $Y_c \cup Y_o$, where $Y_c$ represents closed-set classes in the training set and $Y_o$ represents open-set classes that are not included in the training set, the goal of open-set object recognition is to identify if the image is from a closed-set class or an open-set class. To recognize open-set objects accurately, as suggested by previous works [40, 34], a key challenge is: how to reduce the reliance on *spurious-discriminative* features. To better tackle this challenge, in this work, inspired by that large models can hold very rich while distinct implicit knowledge due to their different training paradigms, we propose a novel open-set object recognition framework **LMC**. Specifically, as shown in Fig. 2, LMC first extracts the implicit knowledge from different off-the-shelf large models to simulate additional virtual open-set classes that share the *spurious-discriminative* features. After that, LMC utilizes these virtual open-set classes to make the *spurious-discriminative* features "less discriminative" during its inference process.

Below, we first describe how LMC extracts implicit knowledge from different large models to effectively simulate virtual open-set classes. After that, we introduce the inference process of LMC.

## 3.1 Virtual Open-set Class Simulation

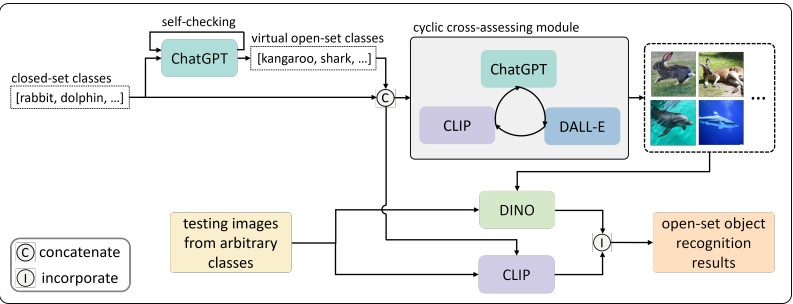

Figure 2: Illustration of the proposed LMC framework.

In the proposed LMC, to reduce the reliance on *spurious-discriminative* features, we aim to utilize the rich implicit knowledge of different large models to simulate additional virtual open-set classes. Specifically, motivated by that knowledge from different modalities (e.g., text and image) can contain different details [60], we aim to leverage implicit knowledge of large models in both the text domain and the image domain to facilitate simulation. To achieve this, we first simulate the names of virtual open-set classes and expand the list of closed-set classes with these virtual open-set classes. We then further generate diverse images for each class in the list to describe each class better.

**Simulating names of virtual open-set classes.** Inspired by that ChatGPT holds rich common-sense knowledge, here we aim to simulate names of virtual open-set classes by asking ChatGPT questions. Specifically, to perform such a simulation in high quality, we find that, it is necessary to 1) ensure that ChatGPT can well understand the question, and 2) encourage ChatGPT to cover as many *spurious-discriminative* features as possible. To achieve these, we introduce the following two designs in our framework.

Inspired by chain-of-thought [59] that large language models can better understand the given instruction when they are guided with intermediate reasoning, to ensure that ChatGPT can well understand the question, we here also guide ChatGPT with intermediate reasoning. Specifically, we find that an effective way to guide ChatGPT is to ask ChatGPT the following three questions for each closed-set class: 1) ``Given a list of classes [classes], can you describe the visual features of each class in the list?'', where [classes] represents the names of the closed-set classes; 2) ``What are the discriminative visual features of class [class]

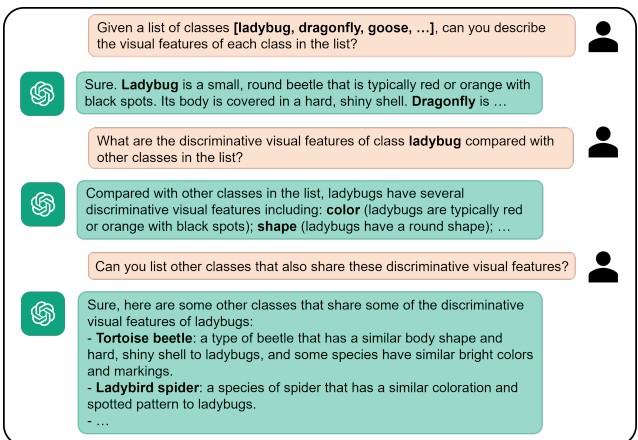

Figure 3: Formulation of language command for ChatGPT in a step-by-step manner with intermediate rationales involved.

compared with other classes in the list?'', where [class] represents the closed-set class that the current set of questions is asked for; 3) ``Can you list other classes that also share these discriminative visual features?''. In Fig. 3, we illustrate an example when the above three questions are asked for the class ladybug. As shown, by formulating the language command in this way, besides the final answer to question 3), ChatGPT is also demanded to generate intermediate rationales that lead to the final answer through the first two questions. This can help ChatGPT to better understand our final purpose, i.e., simulating virtual open-set classes that share *spurious-discriminative* features. In Particular, in Fig. 3, tortoise beetle and ladybird spider are the simulated virtual open-set classes that share *spurious-discriminative* features with the closed-set class ladybug. Through the above three questions, we can simulate the names of virtual open-set classes

that share *spurious-discriminative* features with a single closed-set class. To simulate the names of virtual open-set classes for all closed-set classes, we can simply ask the above three questions for each class in the closed set either sequentially, or concurrently via different ChatGPT instances.

In addition, to simulate a comprehensive list of virtual open-set classes to better cover the *spurious-discriminative* features, we aim to further enable ChatGPT to perform self-checking. In other words, we aim to guide ChatGPT to re-think if there are other *spurious-discriminative* features that are missed. To achieve this, for each closed-set class, after asking the above three questions, we expand the list of classes in question 1) with the obtained virtual open-set classes and re-ask ChatGPT the three questions. Note that by expanding the list with the obtained virtual open-set classes, we exclude the *spurious-discriminative* features that are already covered and thus force ChatGPT to look for *spurious-discriminative* features that are not discovered yet. This iteration of self-checking terminates when ChatGPT stops proposing new virtual open-set classes or a maximum number of cycle times is reached. After performing self-checking, inspired by [34] that all sorts of classes can appear in the real world, we also follow [34] to simulate virtual open-set samples that are less similar to closed-set samples (sharing less *spurious-discriminative* features) via asking ChatGPT: "Given a list of classes [classes], can you name classes that are not similar to them?". Note after expanding the list of closed-set classes with simulated virtual open-set classes through applying the above designs, those *spurious-discriminative* features that are originally discriminative among the closed-set classes would be "less discriminative" in the expanded list.

**Generating diverse images and cross-assessing.** Above we simulate the names of virtual open-set classes $Y_{v\_o}$. Here, inspired by that images can catch different details that can be complementary to the text domain [60], we aim to describe both the closed-set classes and the simulated virtual open-set classes in more detail by further generating diverse images for each class $y \in Y_c \cup Y_{v\_o}$. These generated images will be utilized in the inference process of LMC to reduce its reliance on *spurious-discriminative* features.

Specifically, for each class $y \in Y_c \cup Y_{v\_o}$, we generate diverse images for class $y$ through the following two steps: 1) we first ask ChatGPT to generate diverse text descriptions as: ``Can you write [$K$] diverse descriptions, each describing a different scene about the class [$y$]?'' where $K$ is a hyperparameter representing the number of descriptions generated per class. 2) After that, leveraging the image-generative ability of DALL-E [47], we ask DALL-E to generate images based on these descriptions for each class as:

$$i_y^k = \text{DALL-E}(d_y^k), \textbf{ where } k \in \{1, ..., K\} \tag{1}$$

where $d_y^k$ denotes the $k$-th description generated by ChatGPT for class $y$, and $i_y^k$ denotes the image generated by DALL-E based on $d_y^k$. While we can directly use the images generated above during inference, as shown in Fig. 5 which demonstrates a number of such images, we realize that some of such images may not hold a high enough quality to accurately represent their desired classes (e.g., $i_y^k$ may not represent class $y$ accurately). Thus, directly using the above images can lead to suboptimal model performance during inference.

To address this problem, referring the images that cannot accurately represent their desired classes as the *less accurate* images, we aim to automatically detect these *less accurate* images, refine their corresponding descriptions, and regenerate them accordingly. To achieve this, inspired by that humans can refine their understanding based on others' feedback, here we want to investigate, *can a large model also refine its output based on another large model's feedback?* In particular, we propose a cyclic cross-assessing module, where CLIP acts as the "feedback provider", and provides ChatGPT with feedback about the *less accurate* images to help refine their corresponding descriptions.

Specifically, denote $i_y^k$ an image generated for class $y$, and $D$ the text descriptions "a photo of [class]" for all the classes (i.e., $Y_c \cup Y_{v\_o}$). The cross-assessing module operates by iteratively conducting the following three steps: **1) Detection.** To detect the *less accurate* images, we first leverage the strong image-text alignment ability of CLIP to align $i_y^k$ with $D$ as:

$$p_{assess} = softmax\Big(\text{CLIP}_{vis}(i_y^k)\big(\text{CLIP}_{text}(D)\big)^T\Big) \tag{2}$$

where $\text{CLIP}_{vis}$ denotes the CLIP's visual encoder, and $\text{CLIP}_{text}$ denotes the CLIP's text encoder. The image $i_y^k$ is determined as *less accurate* if compared to class $y$, it is more aligned towards another class $u$ (i.e., instead of class $y$, another class $u$ corresponds to the highest probability value in $p_{assess}$). **2) Refinement.** Next, if $i_y^k$ is determined as a *less accurate* image, in order for its corresponding

description $d_y^k$ to describe class $y$ more properly, we ask ChatGPT to refine $d_y^k$ based on feedback from CLIP as: "This description [$d_y^k$] seems more like class [$u$] to me. Can you refine it to enhance the characteristics of class [$y$]?". **3) Regeneration.** Lastly, the image $i_y^k$ is regenerated through DALL-E based on the refined text description.

Through conducting the above three steps iteratively, during every iteration, we can leverage CLIP to provide specific feedback for ChatGPT so that ChatGPT can refine the descriptions more toward the direction of our desire. With these refined descriptions, images that can well represent each class will be generated. The above iteration terminates when all the generated images are most aligned with their desired classes or a maximum number of cycle times is reached. Note after reaching the maximum number of cycle times, we discard those images that are still *less accurate*.

### 3.2 Inference Process

Above we expand the list of closed-set classes with additional virtual open-set classes, and generate diverse images for each class in the expanded list. In this section, we discuss how our framework utilizes the expanded list and the generated images to make the *spurious-discriminative* features "less discriminative" during its inference process. Note our framework does not require any training process before its inference, which can facilitate its convenient usage in real-world applications.

Specifically, during inference, inspired by the strong image-text alignment ability of CLIP [45] and the strong image-image alignment ability of DINO [5], we propose to adopt both CLIP and DINO to perform open-set object recognition. Note CLIP and DINO are used in collaboration since they can complement each other well. To be more concrete, by aligning the testing image with the names of all the classes in the expanded list (i.e., $Y_c \cup Y_{v\_o}$), we first adopt CLIP to match the testing image with the overall concept of each class. Meanwhile, by leveraging DINO to align the testing image with the generated images of each class in image patch level, we better match the testing image with the detailed local features of each class.

Before entering the inference process, with the notice that certain CLIP and DINO features are used repeatedly across different testing images, we first pre-store these features to keep the efficiency of the inference process. Specifically, denoting $f_{text}^{CLIP} = CLIP_{text}(D)$ the CLIP text feature of all classes, we first store $f_{text}^{CLIP}$, which has already been computed in the above Eq. 2. Moreover, denoting $DINO_{vis}$ the DINO's encoder and $I_y$ the set of images generated from class $y$ through the cyclic cross-assessing module, we compute the DINO visual feature for class $y$ as $f_y^{DINO} = DINO_{vis}(I_y)$ and store $f_y^{DINO}$ for each class $y \in Y_c \cup Y_{v\_o}$.

After pre-storing these features, we then introduce the inference process of the proposed LMC framework. Specifically, given a testing image $i_{test}$, utilizing pre-stored features, LMC first aligns $i_{test}$ with the names of all the classes in $Y_c \cup Y_{v\_o}$ through CLIP as:

$$p_{CLIP} = softmax\big(CLIP_{vis}(i_{test})(f_{text}^{CLIP})^T\big) \qquad (3)$$

where $p_{CLIP}$ denotes the softmax probability derived from CLIP. At the same time, LMC also aligns $i_{test}$ with the generated images through DINO as:

$$p_{DINO} = softmax\big(\{l_{DINO}^y \mid y \in Y_c \cup Y_{v\_o}\}\big), \textbf{ where } l_{DINO}^y = average\big(DINO_{vis}(i_{test})(f_y^{DINO})^T\big) \qquad (4)$$

where $l_{DINO}^y$ is derived from aligning $i_{test}$ with $I_y$ through DINO, and $p_{DINO}$ denotes the softmax probability correspondingly calculated for all the classes in $Y_c \cup Y_{v\_o}$. After deriving $p_{CLIP}$ and $p_{DINO}$, LMC then incorporates them to determine whether $i_{test}$ is from the closed-set classes or open-set classes as:

$$p_{inc} = \alpha p_{CLIP} + (1 - \alpha)p_{DINO}, \quad S = \max_{y \in Y_c}\big(p_{inc}(y|i_{test})\big) \qquad (5)$$

where $\alpha$ is a hyperparameter denoting the incorporation weight, $p_{inc}$ denotes the softmax probability derived from incorporating $p_{CLIP}$ and $p_{DINO}$, $p_{inc}(y|i_{test})$ denotes the probability value that class $y$ corresponds to in $p_{inc}$, and $S \in [0, 1]$ is the closed-set score measured by our framework. Following [34], we represent $S$ as the maximum probability value among the closed-set classes. By performing open-set object recognition in the above way, open-set images would no longer be easily misidentified into the closed-set classes due to the contained *spurious-discriminative* features. This is because, after expanding the list of closed-set classes with the virtual open-set classes, in the expanded list, besides a certain closed-set class, a *spurious-discriminative* feature can be held by some virtual open-set classes as well. Then, when an open-set testing image containing *spurious-discriminative* features is

Table 1: The AUROC results on the detection of closed-set and open-set samples. Results are averaged over five random dataset splits following [6, 56].

| Method | Venue | CIFAR10 | CIFAR+10 | CIFAR+50 | TinyImageNet |
|---|---|---|---|---|---|
| Methods that involve a training process | | | | | |
| OSRCI [36] | ECCV 2018 | 69.9 ± 3.8 | 83.8 ± - | 82.7 ± - | 58.6 ± - |
| C2AE [39] | CVPR 2019 | 89.5 ± - | 95.5 ± - | 93.7 ± - | 74.8 ± - |
| RPL [7] | ECCV 2020 | 90.1 ± - | 97.6 ± - | 96.8 ± - | 80.9 ± - |
| OpenHybrid [68] | ECCV 2020 | 95.0 ± - | 96.2 ± - | 95.5 ± - | 79.3 ± - |
| CVAECapOSR [16] | ICCV 2021 | 83.5 ± 2.3 | 88.8 ± 1.9 | 88.9 ± 1.7 | 71.5 ± 1.8 |
| ARPL [6] | TPAMI 2021 | 90.1 ± 0.5 | 96.5 ± 0.6 | 94.3 ± 0.4 | 76.2 ± 0.5 |
| ARPL+CS [6] | TPAMI 2021 | 91.0 ± 0.7 | 97.1 ± 0.3 | 95.1 ± 0.2 | 78.2 ± 1.3 |
| PMAL [31] | AAAI 2022 | 95.1 ± - | 97.8 ± - | 96.9 ± - | 83.1 ± - |
| ZOC [12] | AAAI 2022 | 93.0 ± 1.7 | 97.8 ± 0.6 | 97.6 ± 0.0 | 84.6 ± 1.0 |
| MLS [56] | ICLR 2022 | 93.6 ± - | 97.9 ± - | 96.5 ± - | 83.0 ± - |
| DIAS [34] | ECCV 2022 | 85.0 ± 2.2 | 92.0 ± 1.1 | 91.6 ± 0.7 | 73.1 ± 1.5 |
| Class-inclusion [8] | ECCV 2022 | 94.8 ± - | 96.1 ± - | 95.7 ± - | 78.5 ± - |
| ODL [30] | TPAMI 2022 | 85.7 ± 1.3 | 89.1 ± 1.4 | 88.3 ± 0.0 | 76.4 ± 1.7 |
| ODL+ [30] | TPAMI 2022 | 88.5 ± 1.3 | 91.1 ± 0.8 | 90.6 ± 0.0 | 74.6 ± 0.8 |
| CSSR [19] | TPAMI 2022 | 91.3 ± - | 96.3 ± - | 96.2 ± - | 82.3 ± - |
| RCSSR [19] | TPAMI 2022 | 91.5 ± - | 96.0 ± - | 96.3 ± - | 81.9 ± - |
| Methods that involve no extra training process | | | | | |
| Softmax | | 93.7 ± 1.7 | 96.5 ± 0.6 | 95.1 ± 1.3 | 83.5 ± 2.7 |
| Ours | | **96.6** ± 0.3 | **98.9** ± 0.7 | **98.5** ± 0.4 | **86.7** ± 1.4 |

Table 2: The OSCR results for open-set object recognition. Results are averaged over five random dataset splits following [6].

| Method | Venue | CIFAR10 | CIFAR+10 | CIFAR+50 | TinyImageNet |
|---|---|---|---|---|---|
| Methods that involve a training process | | | | | |
| GCPL [62] | CVPR 2018 | 84.3 ± 1.7 | 91.0 ± 1.7 | 88.3 ± 1.1 | 59.3 ± 5.3 |
| RPL [7] | ECCV 2020 | 85.2 ± 1.4 | 91.8 ± 1.2 | 89.6 ± 0.9 | 53.2 ± 4.6 |
| ARPL [6] | TPAMI 2021 | 86.6 ± 1.4 | 93.5 ± 0.8 | 91.6 ± 0.4 | 62.3 ± 3.3 |
| ARPL+CS [6] | TPAMI 2021 | 87.9 ± 1.5 | 94.7 ± 0.7 | 92.9 ± 0.3 | 65.9 ± 3.8 |
| ODL [30] | TPAMI 2022 | 84.8 ± 1.4 | 92.5 ± 1.0 | 89.8 ± 0.7 | 64.3 ± 3.2 |
| ODL+ [30] | TPAMI 2022 | 86.9 ± 1.5 | 93.2 ± 0.3 | 90.3 ± 0.2 | 59.2 ± 2.1 |
| Methods that involve no extra training process | | | | | |
| Softmax | | 90.1 ± 1.1 | 94.0 ± 1.4 | 92.8 ± 1.1 | 75.1 ± 5.0 |
| Ours | | **93.6** ± 1.5 | **96.8** ± 0.7 | **96.4** ± 0.4 | **80.6** ± 3.4 |

aligned with the expanded list, it will tend to approach both the closed-set and virtual open-set classes that contain these features at the same time, rather than approaching only certain closed-set classes. Therefore, after softmax normalization, the closed-set score $S$ of the open-set testing image will no longer be easily overestimated due to the *spurious-discriminative* features contained in the image.

# 4 Experiments

To evaluate the effectiveness of our proposed framework LMC, we conduct experiments on four evaluation protocols including CIFAR10, CIFAR+10, CIFAR+50, and TinyImageNet. We conduct our experiments on an RTX 3090 GPU.

## 4.1 Datasets and Evaluation Metric

**CIFAR10.** CIFAR10 [24] is a ten-class object recognition dataset. On this dataset, following [6, 56], 6 closed-set classes and 4 open-set classes are randomly sampled for evaluation each time.

**CIFAR+10 & CIFAR+50.** Following [6, 56], we also evaluate our method on CIFAR+10 and CIFAR+50, which are extensions of the CIFAR10 evaluation protocol. Specifically, following [6, 56], for the CIFAR+$N$ experiments, 4 classes are randomly sampled from CIFAR-10 as the closed-set classes and $N$ non-overlapping classes are randomly sampled from CIFAR-100 [24] as the open-set classes for evaluation ($N = 10$ for CIFAR+10 and $N = 50$ for CIFAR+50).

**TinyImageNet.** TinyImageNet [9] is a subset of ImageNet and contains 200 classes. Following [6, 56], each time, 20 closed-set classes and 180 open-set classes are randomly sampled.

**Evaluation metric.** Following [6, 56], we use the following two metrics: AUROC and OSCR [10]. Among these two metrics, AUROC focuses on evaluating model performance on identifying open-set samples, and OSCR measures the trade-off between classification accuracy and open-set recognition performance. More details especially about the OSCR metric are in the supplementary.

## 4.2 Implementation Details

In LMC, we integrate rich and distinct knowledge from off-the-shelf large models including ChatGPT, DALL-E, CLIP, and DINO in a complementary manner. Specifically, for ChatGPT, we use GPT-3.5-turbo. For CLIP, we use ViT-B/32 for its visual encoder, and the transformer architecture described in [46] for its text encoder. For DINO, we use the second version of DINO [37] with ViT-B/14. We set the maximum number of cycle times for ChatGPT's self-checking to 3. To generate diverse images for each class through the proposed cyclic cross-assessing module, we set the number of diverse detailed descriptions $K$ generated for each class to 10, and the maximum number of cycle times for cyclic cross-assessing to 3. Besides, we set $\alpha$ in incorporating $p_{\mathrm{CLIP}}$ and $p_{\mathrm{DINO}}$ to 0.6.

## 4.3 Experimental Results

We report the AUROC results and the OSCR results respectively in Tab. 1 and Tab. 2. Specifically, besides comparing with existing open-set object recognition methods that assume no access to open-set classes, in these two tables, we also compare our framework with Softmax as a baseline that collaborates large models (CLIP and DINO) and involves no extra training process. Particularly, Softmax does not simulate any virtual open-set classes. It treats the ensemble of CLIP and DINO as a closed-set classifier and computes the maximum probability value of the classifier's output as its score $S$ in the same way as Eq. 3 to Eq. 5 (more details about this baseline are in the supplementary). As shown, compared to state-of-the-art open-set recognition methods and the baseline Softmax, our framework achieves superior performance across different evaluation protocols, demonstrating the effectiveness of our framework.

## 4.4 Ablation Studies

In this section, we conduct extensive ablation experiments on the TinyImageNet evaluation protocol. In particular, in the experiments, we report the AUROC metric averaged over five dataset splits. **More ablation studies such as experiments w.r.t. hyperparameters are in the supplementary.**

**Evaluation on the two alignments performed by LMC.** In LMC, as shown in Fig. 2, during inference, we align the testing image with the images representing each class through DINO, as well as align the testing image with the name of each class through CLIP. Above, we already evaluate our framework as a whole. Here, we further assess the following two variants of our framework, i.e., the variant (**w/o aligning with names**) that only aligns the testing image with the images representing each class, and the variant (**w/o aligning with images**) that only aligns the testing image with the name of each class. As shown in Tab. 3, ignoring either alignment would lead to performance drop compared to our framework. This demonstrates the effectiveness of our framework in concurrently aligning the testing image with the names and the generated images.

Table 3: Evaluation on the two alignments performed by LMC.

| Method | AUROC |
|---|---|
| w/o aligning with names | 84.4 |
| w/o aligning with images | 82.1 |
| LMC | 86.7 |

**Evaluation on the two designs for class name simulation.** To simulate names of virtual open-set classes in high quality, in our framework, we introduce two designs to formulate language commands for ChatGPT properly, i.e., guiding ChatGPT with intermediate reasoning and enabling ChatGPT to perform self-checking.

Table 4: Evaluation on the two designs for class name simulation.

| Method | AUROC |
|---|---|
| w/o intermediate reasoning & self-checking | 84.1 |
| w/o intermediate reasoning | 85.6 |
| w/o self-checking | 85.9 |
| LMC | 86.7 |

Here, to assess the effectiveness of the two designs introduced, we test three variants: **1) w/o intermediate reasoning & self-checking:** this variant uses neither of the two designs and simulates names of virtual open-set classes for each closed-set class via directly asking ChatGPT the following question: ``Given a list of classes [classes], can you name other classes that share visual features with [class] while these shared features are discriminative in the list?'', where [class] represents a closed-set class and [classes]

represents the list of closed-set classes. **2) w/o intermediate reasoning:** this variant enables ChatGPT to perform self-checking but does not guide ChatGPT with intermediate reasoning. Specifically, in this variant, after asking ChatGPT the question in variant **1)**, we expand [classes] in this question with the obtained virtual open-set classes and re-ask ChatGPT this question iteratively until ChatGPT stops proposing new virtual open-set classes or a maximum number of cycle times is reached. As with our framework, we set the maximum number of cycle times to 3 here as well. **3) w/o self-checking:** this variant guides ChatGPT with intermediate reasoning but does not perform self-checking. Specifically, in this variant, we only ask the three questions proposed in Sec. 3.1 once but not in an iterative manner. As shown in Tab. 4, our proposed framework consistently outperforms all three variants. This demonstrates (1) the effectiveness of the intermediate reasoning design, which can lead ChatGPT to better understand the asked question; (2) the effectiveness of the self-checking design, which can lead ChatGPT to simulate a more comprehensive list of virtual open-set classes.

**Impact of the cyclic cross-assessing module.** To ensure that the images generated by DALL-E can well represent their desired classes, in our framework, we propose a cyclic cross-assessing module to refine the generated images iteratively. To validate this module, we consider three alternative ways to generate images: **1) w/o cross-assessing**: this variant directly passes the generated images to the inference process of our framework without checking or refining them. **2) Check and discard**: this variant

Table 5: Evaluation on the effectiveness of the cyclic cross-assessing module.

| Method | AUROC |
|---|---|
| w/o cross-assessing | 84.5 |
| Check and discard | 84.9 |
| Check and naively refine | 85.4 |
| LMC | 86.7 |

checks the generated images through CLIP and passes only those images that pass the checking (i.e., those images that are most aligned towards their desired classes) to the inference process of our framework. Note in this variant, those images that do not pass the checking are simply discarded and no refinement is involved. **3) Check and naively refine**: this variant both checks and refines the generated images. However, in this variant, we directly ask ChatGPT to refine the descriptions of those incorrect images, and the feedback from CLIP (i.e., a target refinement direction) is not involved in the refinement. As shown in Tab. 5, the performance of all three variants are worse than our method. This shows the effectiveness of our proposed cyclic cross-assessing module, which can leverage CLIP to provide specific feedback for ChatGPT, so that ChatGPT can refine the descriptions more toward the direction of our desire.

**Overlap between the virtual open-set classes and the open-set classes used for evaluation.** In our method, we simulate virtual open-set classes to lead the framework to less rely on *spurious-discriminative* features. Here, we aim to verify that the effectiveness of our framework does not come from just matching the open-set classes used for evaluation. To achieve this, we evaluate it on two subsets

Table 6: Analysis on the overlap between the virtual open-set classes and the open-set classes used for evaluation.

| Method | AUROC | |
|---|---|---|
| | Subset A | Subset B |
| Baseline(Softmax) | 83.5 | 83.6 |
| LMC | 93.1 | 86.5 |

of the open-set classes used for evaluation. Specifically, the first subset (**subset A**) includes all those open-set classes that are contained in the list of simulated virtual open-set classes $Y_{v\_o}$, and the second subset (**subset B**) includes all the other open-set classes that are not contained in $Y_{v\_o}$. As shown in Tab. 6, compared to Softmax as a baseline, even for subset B, there is still a significant performance improvement, demonstrating the effectiveness of our method even when $Y_{v\_o}$ and $Y_o$ do not overlap at all. Moreover, we would like to point out that the virtual open-set classes and the open-set classes

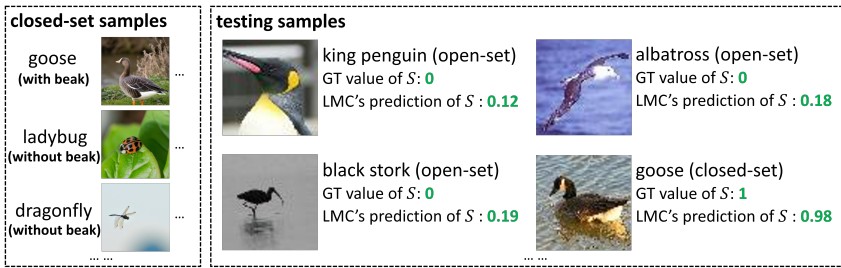

Figure 4: Qualitative results on the TinyImageNet dataset. As shown, even when the testing image holds the *spurious-discriminative* feature ("with beak" here), our framework can accurately identify if it is from closed-set or open-set classes. The above results are based on the fourth dataset split following [6]. More qualitative results are in the supplementary.

used for evaluation barely overlap. For example, for each data split of TinyImageNet, out of its 180 open-set classes used for evaluation, only no more than 6 classes (i.e., no more than $3.4\%$ of classes) are within the list of virtual open-set classes. Yet, our method still achieves state-of-the-art performance. This further demonstrates that the effectiveness of our method does not naively come from matching the real open-set classes.

**Qualitative results.** Some qualitative results are shown in Fig. 4. As shown, our framework can perform open-set object recognition accurately even when the testing images contain *spurious-discriminative* features. This demonstrates that our framework can effectively reduce the reliance on *spurious-discriminative* features when performing open-set object recognition.

**Visualization of images re-generated through the cross-assessing module.** In Fig. 5, we visualize some images generated before and after passing through the proposed cross-assessing module. As shown, the images before regeneration cannot well represent their desired classes, whereas the images regenerated through the cross-assessing module can represent their desired classes more accurately. This demonstrates the effectiveness of the proposed cross-assessing module.

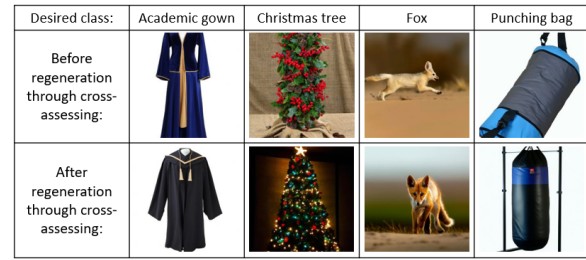

Figure 5: Visualization of the images generated before and after passing through the cross-assessing module. More visualizations are in the supplementary.

## 5 Conclusion

In this paper, we have proposed a novel open-set object recognition framework LMC, which collaborates different large models to recognize open-set objects. Specifically, in LMC, we utilize rich and distinct knowledge from different off-the-shelf large models to reduce the reliance of the proposed framework on *spurious-discriminative* features. Moreover, we also incorporate our framework with several designs to effectively extract implicit knowledge from large models. Our framework achieves state-of-the-art performance on four widely used evaluation protocols in a training-free manner. Besides, our method may potentially also (1) be applied in other related tasks such as confidence estimation [44, 43] and uncertainty estimation [25, 29]; (2) inspire new ways for follow-up research. Below, we list a few possible new ways that follow-up research work can further investigate: (1) how to collaborate large models with conventional open-set object recognition methods to further improve performance; (2) how to facilitate the usage of black-box large models in our framework through black-box optimization algorithms such as the CMA Evolution Strategy [52]. We leave these as our future works.

## Acknowledgments and Disclosure of Funding

This work was supported by the National Research Foundation Singapore under the AI Singapore Programme (Award Number: AISG-100E-2023-121).

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
