# LMC: Large Model Collaboration with Cross-assessment for Training-Free Open-Set Object Recognition
## (Supplementary Material)

**Haoxuan Qu** *
SUTD
Singapore
haoxuan_qu@mymail.sutd.edu.sg

**Xiaofei Hui** *
SUTD
Singapore
xiaofei_hui@sutd.edu.sg

**Yujun Cai**
Meta
U.S.
yujuncai@meta.com

**Jun Liu** †
SUTD
Singapore
jun_liu@sutd.edu.sg

## 1 Additional Qualitative Results

In Fig. 1, we compare our LMC framework with the baseline Softmax, and present qualitative results on the TinyImageNet dataset. Note that for the baseline Softmax, we do not simulate any virtual open-set classes. As shown, via simulating additional virtual open-set classes that share the *spurious-discriminative* features, our framework can prevent the closed-set score $S$ of the open-set testing image from being easily overestimated by approaching the image to both a certain closed-set class and certain virtual open-set classes. This demonstrates the effectiveness of our framework in reducing the reliance on *spurious-discriminative* features.

## 2 Details of Our Used Evaluation Metrics

In our experiments, following [1, 11], we use the following two metrics: AUROC and OSCR [3]. Below, we discuss them in more detail.

**AUROC.** AUROC is a widely-used threshold-independent evaluation metric. It measures the area of the region under the Receiver Operating Characteristic (ROC) curve. We directly use the function inside the scikit-learn package to calculate AUROC.

**OSCR.** OSCR, introduced in [3], measures the trade-off between classification accuracy and open-set recognition performance. Let $\theta \in [0, 1]$ be a threshold. According to [3, 1], to calculate OSCR, we first need to calculate the Correct Classification Rate (CCR) and the False Positive Rate (FPR) at $\theta$. Specifically, denoting $D_c$ testing images from closed-set classes, CCR is calculated as the fraction of images from $D_c$ where the correct class $\hat{y}$ has maximum probability and this probability is greater than or equal to $\theta$:

$$\text{CCR}(\theta) = \frac{\left| \{ i_{test} \mid \left( i_{test} \in D_c \right) \wedge \left( \underset{y \in Y_c}{\arg\max} \, Prob(y \mid i_{test}) = \hat{y} \right) \wedge \left( \frac{Prob(\hat{y} \mid i_{test})}{\sum_{y \in Y_c} Prob(y \mid i_{test})} \geq \theta \right) \} \right|}{|D_c|}$$

(1)

---

*Both authors contributed equally to the work.

†Corresponding Author

37th Conference on Neural Information Processing Systems (NeurIPS 2023).

Moreover, denoting $D_o$ testing images from open-set classes, FPR is calculated as the fraction of images from $D_o$ that are misclassified as any closed-set class $y \in Y_c$ with a probability greater than or equal to $\theta$:

$$\text{FPR}(\theta) = \frac{\left| \left\{ i_{test} \mid \left( i_{test} \in D_o \right) \wedge \left( \frac{\max_{y \in Y_c} Prob(y \mid i_{test})}{\sum_{y \in Y_c} Prob(y \mid i_{test})} \geq \theta \right) \right\} \right|}{|D_o|} \tag{2}$$

After calculating CCR and FPR, we then plot CCR versus FPR at different $\theta$. OSCR is then calculated as the area under the plotted curve.

## 3 Details of Baseline Softmax

In our main paper, we introduce a baseline named Softmax, which treats the ensemble of CLIP and DINO as a closed-set classifier, and does not simulate any virtual open-set classes. Below, we introduce this baseline in more detail.

Before entering the inference process, similar to our framework, Softmax also pre-stores certain CLIP and DINO features to make the inference process more efficient. Denote $D_{Softmax}$ the text descriptions "a photo of [class]" for all the closed-set classes (i.e., $Y_c$), $\text{CLIP}_{text}$ the CLIP's text encoder, and $f_{Softmax}^{\text{CLIP}} = \text{CLIP}_{text}(D_{Softmax})$ the CLIP text feature of all the closed-set classes, Softmax first stores $f_{Softmax}^{\text{CLIP}}$. Moreover, denote $\text{DINO}_{vis}$ the DINO's encoder, and $I_y$ the set of images generated from class $y$ through the cyclic cross-assessing module. Softmax also stores the DINO visual feature $f_y^{\text{DINO}} = \text{DINO}_{vis}(I_y)$ for each closed-set class $y \in Y_c$.

Then during the inference process, given a testing image $i_{test}$, Softmax first aligns $i_{test}$ with the names of the closed-set classes through CLIP as:

$$p_{\text{CLIP}}^{Softmax} = softmax\left(\text{CLIP}_{vis}(i_{test})(f_{softmax}^{\text{CLIP}})^T\right) \tag{3}$$

where $\text{CLIP}_{vis}$ denotes the CLIP's visual encoder and $p_{\text{CLIP}}^{Softmax}$ denotes the softmax probability derived from CLIP in baseline Softmax. At the same time, Softmax also aligns $i_{test}$ with the images generated from the closed-set classes through DINO as:

$$p_{\text{DINO}}^{Softmax} = softmax\left(\{l_{\text{DINO}}^y \mid y \in Y_c\}\right), \textbf{ where } l_{\text{DINO}}^y = average\left(\text{DINO}_{vis}(i_{test})(f_y^{\text{DINO}})^T\right) \tag{4}$$

where $l_{\text{DINO}}^y$ is derived from aligning $i_{test}$ with $I_y$ through DINO, and $p_{\text{DINO}}^{Softmax}$ denotes the softmax probability correspondingly calculated for all the closed-set classes. After deriving $p_{\text{CLIP}}^{Softmax}$ and $p_{\text{DINO}}^{Softmax}$, Softmax then incorporates them to calculate the closed-set score $S$ as:

$$p_{inc}^{Softmax} = \alpha p_{\text{CLIP}}^{Softmax} + (1 - \alpha) p_{\text{DINO}}^{Softmax}, \quad S = \max_{y \in Y_c}\left(p_{inc}^{Softmax}(y|i_{test})\right) \tag{5}$$

where $\alpha$ is a hyperparameter denoting the incorporation weight, $p_{inc}^{Softmax}$ denotes the softmax probability derived from incorporating $p_{\text{CLIP}}^{Softmax}$ and $p_{\text{DINO}}^{Softmax}$, and $p_{inc}^{Softmax}(y|i_{test})$ denotes the probability value that class $y$ corresponds to in $p_{inc}^{Softmax}$.

## 4 Additional Ablation Studies

Here, we conduct more ablation experiments. In these experiments, unless specifically specified, we report the AUROC metric averaged over five dataset splits on the TinyImageNet evaluation protocol.

**Experiments under the cross-dataset setup.** In our main paper, we evaluate our framework under the standard set-up where the open-set class images and the closed-set class images come from the same dataset. Here, we also evaluate our method under a common cross-dataset setup following [12, 9, 10, 14, 5]. In this setup, closed-set class images are from CIFAR-10, and open-set class images are from ImageNet-crop [2], ImageNet-resize [2], LSUN-crop [13], and LSUN-resize [13]. We use the same evaluation metric F1-score as [12, 9, 10, 14, 5] under this setup. As shown in Tab. 1, under the cross-dataset setup, our method also achieves significant performance improvement over previous methods, further demonstrating the effectiveness of our method.

Table 1: Experiments under the cross-dataset setup.

| Method | ImageNet-crop | ImageNet-resize | LSUN-crop | LSUN-resize |
|---|---|---|---|---|
| CROSR [12] | 73.3 | 76.3 | 72.0 | 74.9 |
| GFROSR [9] | 82.1 | 79.2 | 84.3 | 80.5 |
| CGDL [10] | 84.0 | 83.2 | 80.6 | 81.2 |
| PROSER [14] | 84.9 | 82.4 | 86.7 | 85.6 |
| CVAECapOSR [5] | 85.7 | 83.4 | 86.8 | 88.2 |
| ZOC [4] | 84.6 | 81.8 | 87.4 | 86.8 |
| PMAL [7] | 85.8 | 83.2 | 86.5 | 87.6 |
| LMC | **88.0** | **86.0** | **91.5** | **93.5** |

**Impact of using different backbones for the CLIP's visual encoder.** In the experiments in the main paper, we leverage CLIP and DINO in collaboration to perform open-set object recognition during inference. Specifically, we use CLIP with ViT-B/32 as the backbone for its visual encoder, and we use the second version of DINO with ViT-B/14 as its backbone. Here to verify the generality of our framework, we first fix the backbone of DINO and test using other off-the-shelf backbones for the CLIP's visual encoder, including ResNet-101, ViT-B/16, and ViT-L/14. As shown in Tab. 2, our framework with these models used as the backbone for CLIP's visual encoder can also achieve superior performance compared to the state-of-the-art method ZOC [4], demonstrating the generality of our framework.

Table 2: Evaluation on using different backbones for the CLIP's visual encoder.

| Backbone | AUROC |
|---|---|
| ZOC [4] | 84.6 |
| LMC(ResNet-101) | 85.5 |
| LMC(ViT-B/32) | 86.7 |
| LMC(ViT-B/16) | 87.5 |
| LMC(ViT-L/14) | 89.5 |

**Impact of using different backbones for DINO.** Moreover, we also fix the backbone of CLIP and test using other off-the-shelf backbones for the second version of DINO, including ViT-S/14, ViT-L/14, and ViT-G/14. As shown in Tab. 3, our framework with these models used as the backbone for DINO can also achieve better performance compared to the state-of-the-art method ZOC [4]. This further demonstrates the generality of our framework.

Table 3: Evaluation on using different backbones for DINO.

| Backbone | AUROC |
|---|---|
| ZOC [4] | 84.6 |
| LMC(ViT-S/14) | 85.3 |
| LMC(ViT-B/14) | 86.7 |
| LMC(ViT-L/14) | 90.3 |
| LMC(ViT-G/14) | 91.2 |

**Impact of using different versions of GPT.** Besides, we also test using different versions of GPT with different capabilities, including GPT 3.0 and GPT 3.5 (ChatGPT). As shown in Tab. 4, our framework with different versions of GPT used can achieve different results. This shows that the performance of our framework is affected by the version of GPT.

Table 4: Evaluation on using different versions of GPT.

| Backbone | AUROC |
|---|---|
| GPT 3.0 | 85.2 |
| GPT 3.5 | 86.7 |

**Impact of the maximum number of cycle times for ChatGPT's self-checking.** In our framework, we set the maximum number of cycle times for ChatGPT's self-checking to 3. Here we evaluate other choices of maximum cycle times and report the results in Tab. 5. Note to avoid confusion, in our paper, the number of cycle times for self-checking refers to the number of times that the set of three questions is asked. As shown, when the maximum number of cycle times is smaller than 3, the performance of the framework improves with the increase in the maximum number of cycle times. This might be because, by setting the maximum number of cycle times to be a larger number, ChatGPT can better cover the *spurious-discriminative* features. Moreover, when the maximum number of cycle times is larger than 3, the performance does not enhance anymore. Thus, taking the efficiency of the framework into consideration, we set the maximum number of cycle times to 3.

Table 5: Evaluation on the maximum number of cycle times for Chat-GPT's self-checking.

| Method | AUROC |
|---|---|
| Maximum number of cycle times for self-checking = 1 | 85.9 |
| Maximum number of cycle times for self-checking = 2 | 86.4 |
| Maximum number of cycle times for self-checking = 3 | 86.7 |
| Maximum number of cycle times for self-checking = 4 | 86.7 |

**Check w.r.t. the reasonability of the generated descriptions.** In our framework, we use ChatGPT to generate text descriptions for the evaluated classes, and we further design a cyclic cross-assessing module in which we guide ChatGPT to modify its generated descriptions. Here, we perform a check w.r.t. the reasonability of the generated descriptions before and after passing into the cross-assessing

module. Specifically, we find that, before passing into the cross-assessing module, 3% of descriptions are checked to be unreasonable, but none of the descriptions output by the module is checked to be unreasonable. This shows that ChatGPT has small probability to generate unreasonable descriptions, while our cross-assessing module can further mitigate this problem. The above check is done by inviting 3 volunteers and passing the same 1000 descriptions to them. The 3 volunteers first make decisions independently and then discuss disagreed decisions.

**Impact of the number of descriptions $K$.** In our framework, during generating images, we set the number of diverse detailed descriptions $K$ generated for each class to 10. As shown in Tab. 6, the model performance consistently improves with the increase of description numbers. This might be because, with more descriptions per class, the framework can deepen its understanding of all the classes. When the number of descriptions $K$ per class exceeds 10, the performance of the framework becomes stable. Therefore, we set $K$ to 10 in our experiments.

Table 6: Evaluation on the number of descriptions $K$.

| Method | AUROC |
|---|---|
| $K = 1$ | 83.5 |
| $K = 5$ | 86.2 |
| $K = 10$ | 86.7 |
| $K = 15$ | 86.7 |

**Impact of the maximum number of cycle times for cyclic cross-assessing.** In the cyclic cross-assessing module of our framework, we set the maximum number of cycle times to 3. Here we evaluate other choices of

Table 7: Evaluation on the maximum number of cycle times for cyclic cross-assessing.

| Method | AUROC |
|---|---|
| Maximum number of cycle times for cross-assessing = 1 | 84.9 |
| Maximum number of cycle times for cross-assessing = 2 | 86.2 |
| Maximum number of cycle times for cross-assessing = 3 | 86.7 |
| Maximum number of cycle times for cross-assessing = 4 | 86.8 |

maximum cycle times for cyclic cross-assessing and report the results in Tab. 7. As shown, the performance of the framework improves with the increase in the maximum number of cycle times for cyclic cross-assessing. This might be because, by setting the maximum number of cycle times for cyclic cross-assessing to be a larger number, the generated images can be better refined. Moreover, the performance improvement becomes trivial when the maximum number of cycle times for cyclic cross-assessing is larger than 3. Hence, we set the maximum number of cycle times for cyclic cross-assessing to 3.

**Impact of the incorporation weight $\alpha$.** To calculate the closed-set score $S$ during inference, in our framework, we incorporate $p_{\text{CLIP}}$ and $p_{\text{DINO}}$ with $\alpha$ as the incorporation weight. In our experiments, we set $\alpha$ to 0.6. Here we also assess the other choices of $\alpha$ and plot the results in Fig. 2. As shown, with different choices of $\alpha$, the performance of our framework is consistent, which demonstrates the robustness of our framework to this hyperparameter.

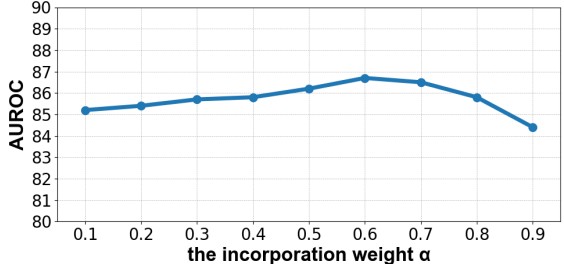

Figure 2: Evaluation of hyperparameter $\alpha$.

**Impact of using the training data.** In the main paper, we have shown the effectiveness of our framework LMC without using any training data (**LMC with generated images only**). Here, to explore the impact of the training data on our framework, we also

Table 8: Evaluation on the usage of the training data.

| Method | AUROC |
|---|---|
| LMC with generated images only | 86.7 |
| LMC with training data and generated images | 87.5 |

assess a variant (**LMC with training data and generated images**). This variant follows the same inference process as our framework, except that besides the generated images, it passes the same number of training images to DINO as well. As shown in Tab. 8, while our framework is already effective without using any training data, using training data can further improve the performance of our framework. This demonstrates that our framework can be used effectively no matter whether the training data is available or not. Also note that in the real world, the training data may no longer be accessible during deployment (e.g., due to privacy concerns [8, 6]). Our framework can better facilitate real-world applications, as it can perform open-set object recognition effectively even without access to training data.

**Evaluation of other baselines.** In our framework, we leverage different off-the-shelf large models in collaboration. To validate the effectiveness of collaborating different large models, we test the following two baselines. In each of these two baselines, only one large model is used. **1) DINO only:** this baseline uses only DINO but no other large models. Specifically, since this baseline does not hold the ability to generate any images, it leverages training images instead. Denote $I_y^t$ the set of training images randomly selected from class $y$ where $|I_y^t| = |I_y|$ for fair comparison. This baseline first calculates $p_{\text{DINO}}^{Softmax}$ in the same way as in Eq. 4 above, except that $f_y^{\text{DINO}}$ is calculate from $I_y^t$ but not $I_y$. This baseline then calculates the closed-set score $S$ as $S = \max_{y \in Y_c} \left( p_{\text{DINO}}^{Softmax}(y|i_{test}) \right)$. **2) CLIP only:** this baseline uses only CLIP but no other large models to perform open-set object recognition. Specifically, it first calculates $p_{\text{CLIP}}^{Softmax}$ in the same way as in Eq. 3 above. After that, the baseline calculates the closed-set score $S$ as $S = \max_{y \in Y_c} \left( p_{\text{CLIP}}^{Softmax}(y|i_{test}) \right)$. As shown in Tab. 9, both these two baselines utilizing only a single large model cannot perform open-set object recognition well.

Table 9: Evaluation of two other baselines.

| Method | AUROC |
|---|---|
| DINO only | 82.9 |
| CLIP only | 81.1 |
| LMC | 86.7 |

**Impact of specifying the number of simulated classes.** In our framework, during asking ChatGPT questions to simulate the names of virtual open-set classes, we don't specify how many classes ChatGPT should answer and let ChatGPT decide this by itself. Here, we consider an alternative way, in which the number of classes ChatGPT should answer is specified in the asked question. Specifically, in this alternative way, we replace the original third question ``Can you list other classes that also share these discriminative visual features?'' with ``Can you list [$N$] other classes that also share these discriminative visual features?'', where $N$ represents the number of classes this question requires ChatGPT to generate. We report the results with different $N$ in Tab. 10. As shown, this alternative way consistently performs slightly worse than our framework. This might be because, the optimal number of virtual open-set classes for ChatGPT to generate per closed-set class can vary across different closed-set classes. In our framework, we then do not specify the number of simulated classes and let ChatGPT decide this automatically by itself.

Table 10: Evaluation of whether to specify the number of simulated classes.

| Method | AUROC |
|---|---|
| N = 1 | 86.2 |
| N = 2 | 86.5 |
| N = 3 | 86.4 |
| N = 4 | 86.4 |
| LMC | 86.7 |

**Preparation time.** In our framework, before entering the inference process, we first prepare the framework by simulating the names of virtual open-set classes, generating diverse images for each class, and pre-storing certain features of CLIP and DINO. We here report the preparation time of our framework on one dataset split of the TinyImageNet protocol. First, the simulation of the names of virtual open-set classes takes around 5.2 minutes. Next, the generation of diverse images along with cross-assessing takes around 32.7 minutes. Lastly, the pre-storage of CLIP and DINO features takes around 1.0 minute. In this way, our framework totally needs around 38.9 minutes (see Tab. 11) to prepare for one dataset split of the TinyImageNet protocol. Note that a dataset split from one of the other three evaluation protocols, with fewer closed-set classes, can have even shorter preparation time. Also note that the whole preparation process of our framework can be achieved with a single script automatically.

Table 11: Total preparation time of our framework. Note that the whole preparation process can be automatically conducted with a script.

| Method | Total preparation time |
|---|---|
| LMC | around 38.9 min |

**Inference speed.** We test the inference speed of our framework on an RTX 3090 GPU. Based on the test, our framework can process around 45.2 images per second (see Tab. 12), which satisfies most real-time requirements.

Table 12: Inference speed of our framework.

| Method | Inference speed |
|---|---|
| LMC | around 45.2 images per sec |

# 5 Additional Visualizations

**Visualization of some dialogues between ChatGPT and us.** In Fig. 3, we visualize some dialogues between ChatGPT and us. Note in these dialogues, ChatGPT is guided with intermediate reasoning through our asked questions. As shown, when guided with intermediate reasoning in this way, ChatGPT can understand our purpose, and simulate virtual open-set classes that share *spurious-discriminative* features successfully.

**Visualization w.r.t. self-checking.** In our framework, we propose to enable ChatGPT to perform self-checking, so that a more comprehensive list of virtual open-set classes can be simulated to better cover the *spurious-discriminative* features. In Fig. 4, we visualize the self-checking process. As shown, when enabled to perform self-checking, ChatGPT can better cover the *spurious-discriminative* features, and simulate names of virtual open-set classes in a more comprehensive manner.

**Visualization w.r.t. the cyclic cross-assessing module.** In this work, we also incorporate our framework with a cyclic cross-assessing module, which is designed to refine those generated images that are *less accurate*. In Fig. 5, we visualize the refining process of the proposed cyclic cross-assessing module. As shown, through the proposed module, those images that are originally not most aligned with their desired classes can be refined to better represent their desired classes. This demonstrates the effectiveness of the proposed cross-assessing module.

**Visualization of the simulated virtual open-set classes for all the closed-set classes.** Moreover, in Fig. 6, we also visualize the virtual open-set classes simulated for all the closed-set classes in a certain dataset split. As shown, via guiding ChatGPT with intermediate reasoning and enabling ChatGPT to perform self-checking, our proposed framework can effectively simulate virtual open-set classes for all the closed-set classes.

# 6 Licenses

**Dataset licenses.** We use CIFAR10 dataset by following the MIT License. We use CIFAR100 dataset in CIFAR+10 and CIFAR+50 evaluation protocols by following the MIT License. As for the TinyImageNet dataset, as it is a subset of ImageNet, we follow the license of ImageNet.

**Large model licenses.** We use ChatGPT and DALL-E by following the terms of using the services of OpenAI. We use CLIP by following the MIT License. We use DINO by following this license.

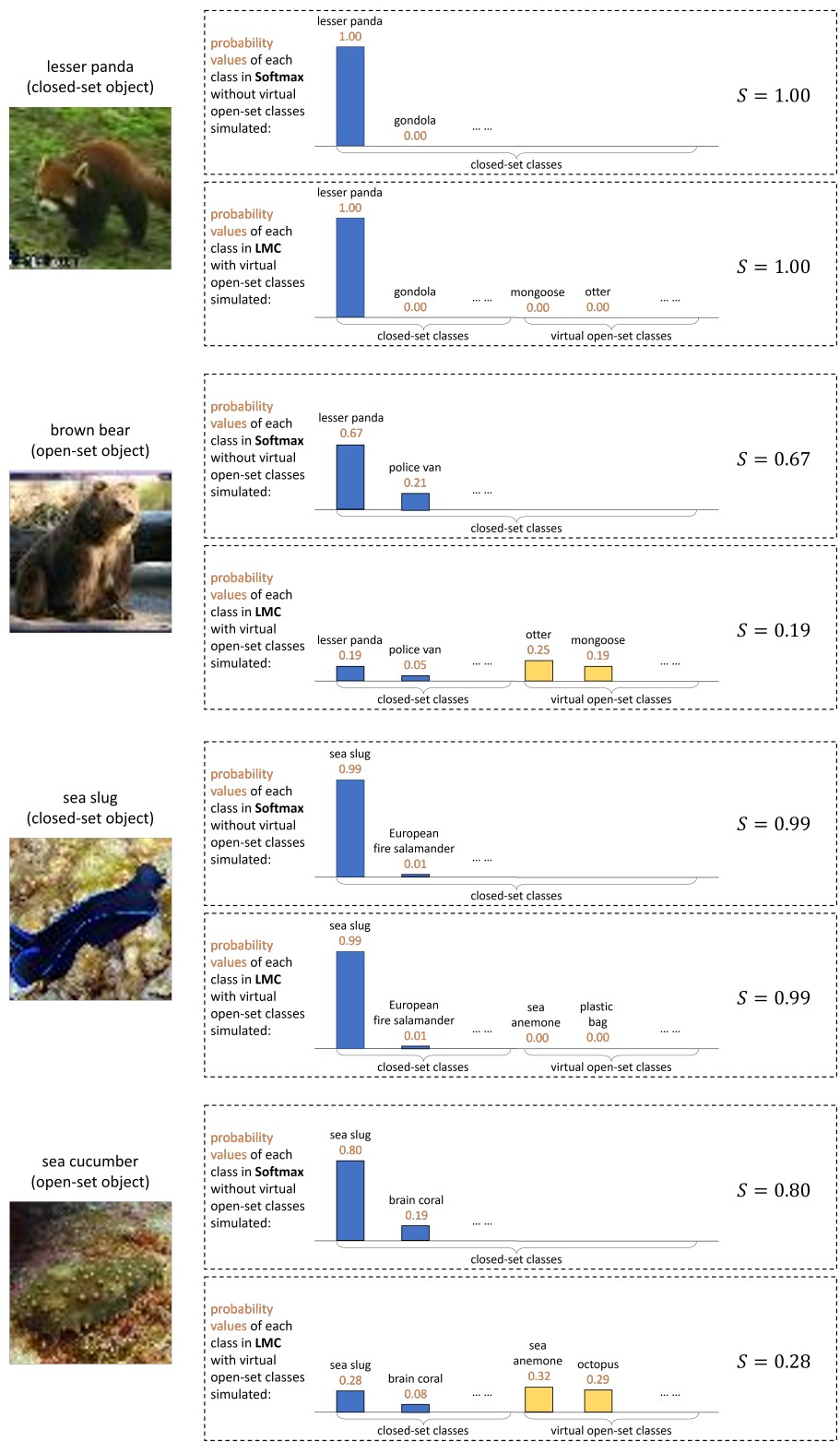

Figure 1: Qualitative results of our proposed LMC framework and the baseline Softmax. The above results are based on the first dataset split following [1].

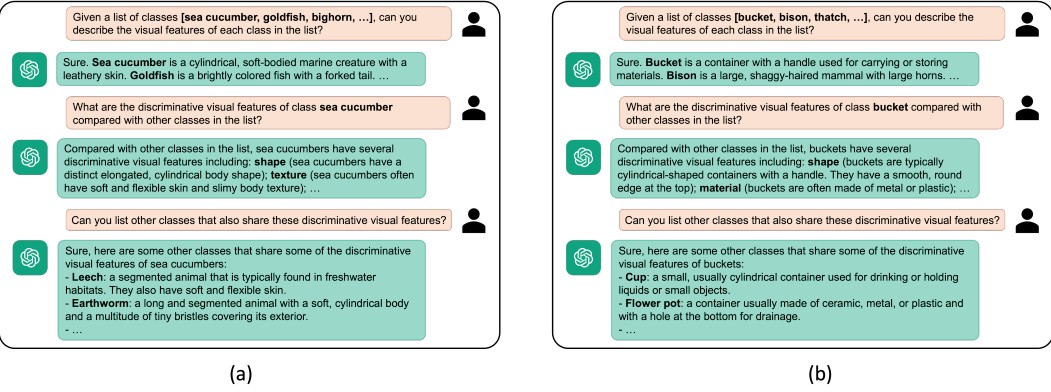

Figure 3: Visualization of some dialogues between ChatGPT and us.

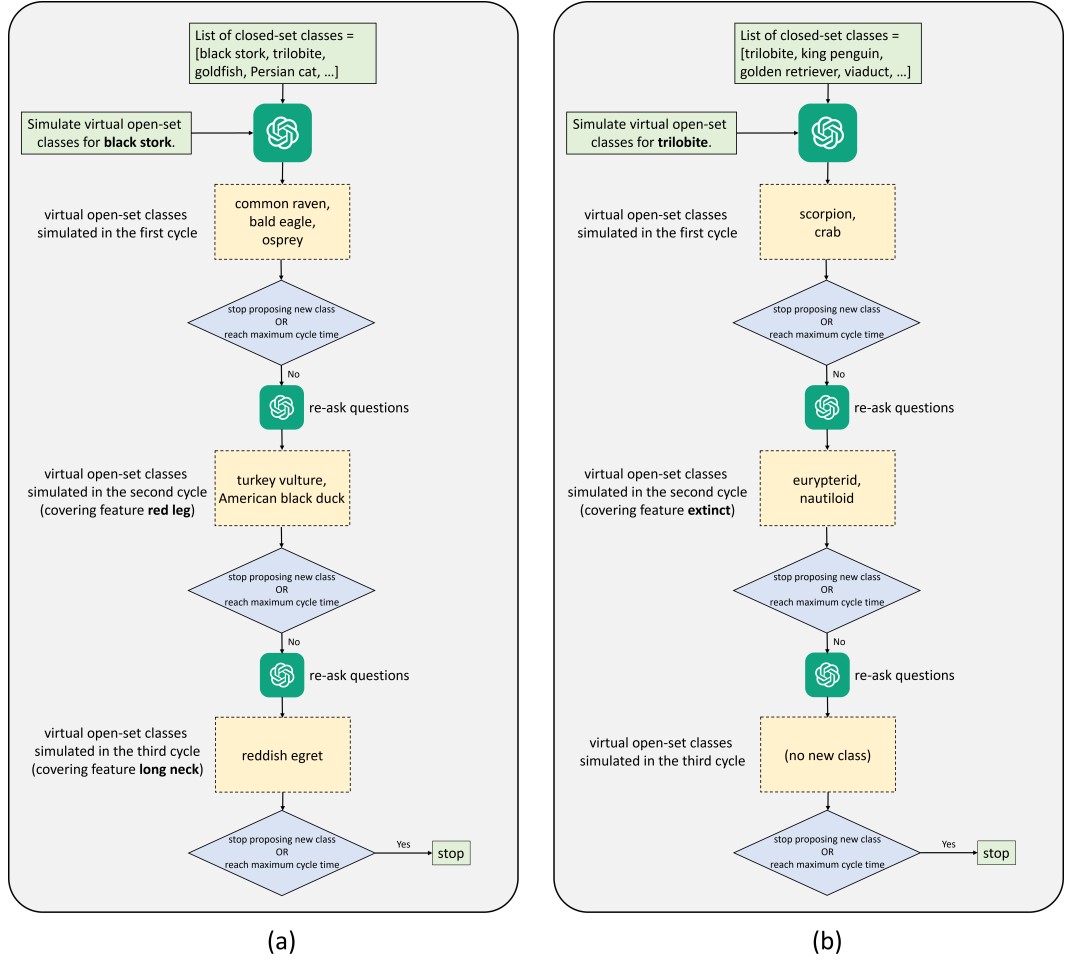

Figure 4: Visualization w.r.t. self-checking. In (a), self-checking terminates when a maximum number of cycle times is reached. In (b), self-checking terminates when ChatGPT stop proposing new virtual open-set classes. As shown, when ChatGPT is enabled to perform self-checking, it can better cover the *spurious-discriminative* features, and more comprehensively simulate names of virtual open-set classes.

| Desired class: | Chameleon | | Ambulance | |
|---|---|---|---|---|
| | Description | Image | Description | Image |
| The first cycle of cross-assessing: | A camouflaging chameleon with small grasping digits on its feet to cling to tree bark and evade danger. |  | A military-grade ambulance with sturdy metal plating and heavy-duty wheels |  |
| The second cycle of cross-assessing: | A chameleon perched on a branch, slowly changes its skin color to blend in with the surroundings, scanning the environment for prey or predators. |  | A robust ambulance built for emergencies, equipped with durable metal plating and reliable heavy-duty wheels to swiftly navigate any terrain. |  |
| The third cycle of cross-assessing: | A chameleon clings to a twig, its skin morphing into an intricate pattern of green, yellow, and brown. Its large, expressive eyes follow every movement with precision. |  | A lifesaving ambulance stands poised, complete with emergency lights and medical equipment neatly stowed away for quick access. |  |

(a)

| Desired class: | Black stork | | Scarf | |
|---|---|---|---|---|
| | Description | Image | Description | Image |
| The first cycle of cross-assessing: | From a distance, a black stork appears almost regal as it glides gracefully through the clouds, high above the treetops below. |  | The cozy red scarf showcases delightful white snowflake patterns, adding a touch of winter charm. |  |
| The second cycle of cross-assessing: | With its sleek black feathers and long, pointed beak, the black stork cuts a distinct figure in the sky, effortlessly riding the thermals and displaying its incredible wingspan. |  | The red scarf exudes warmth with its cozy texture and features elegant white snowflake patterns, featuring subtle fringing along the edges. |  |

(b)

Figure 5: Visualization w.r.t. the cyclic cross-assessing module. Images in (a) are not most aligned with their desired classes until the third cycle, and images in (b) are not most aligned with their desired classes until the second cycle.

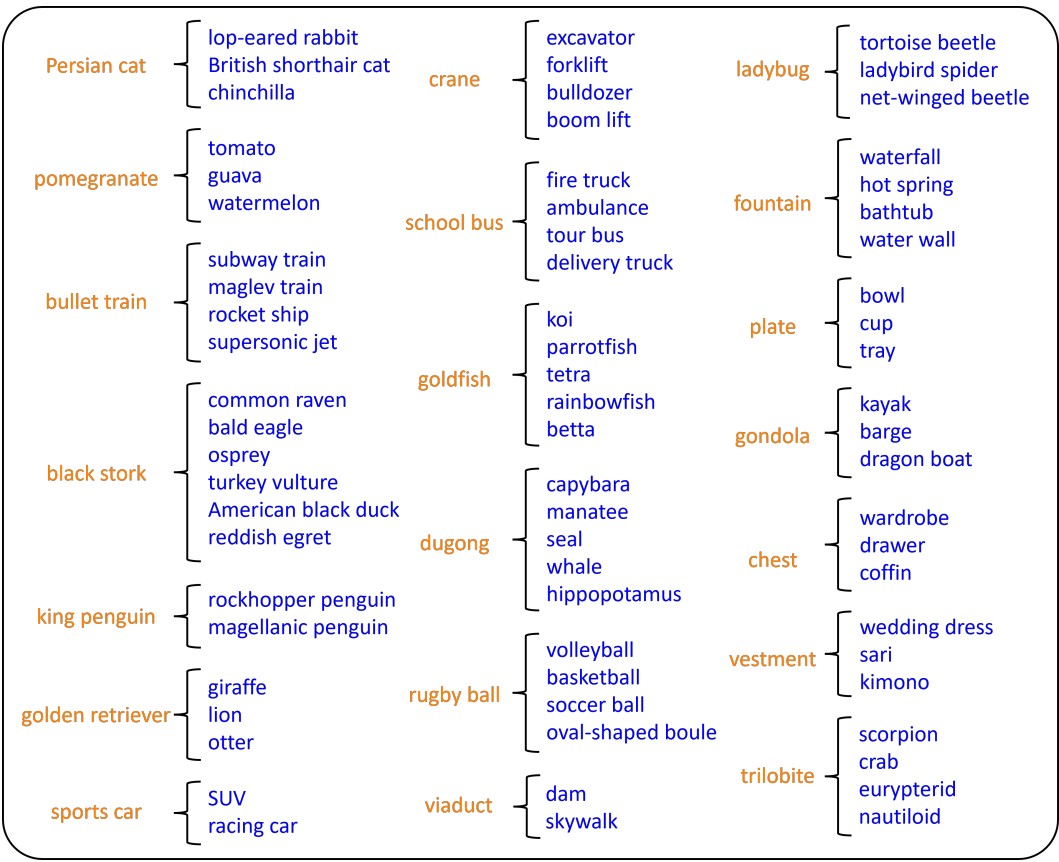

Figure 6: Visualization of the simulated virtual open-set classes for all the closed-set classes for the fifth dataset split following [1]. Here, orange text represents the closed-set classes and blue text represents the corresponding simulated virtual open-set classes.