# OpenReview forum: "LMC: Large Model Collaboration with Cross-assessment for Training-Free Open-Set Object Recognition"
_NeurIPS.cc/2023/Conference — NeurIPS 2023 poster_

### Official Review · Reviewer_zWHw · 2023-07-04

**Soundness:** 3 good
**Presentation:** 3 good
**Contribution:** 2 fair
**Rating:** 6
**Confidence:** 4

**Summary:**

In this paper, the authors tackle the problem of open-set object recognition. The key challenge in open-set recognition problem they tackle is the model's reliance on spurious-discriminative features. To tackle this challenge, they propose an open-set object recognition system consisting of multiple pre-trained foundation models. They employ ChatGPT to generate virtual open-set categories that share spurious-discriminative features with closed-set categories. Then they generate images of the both closed-set and open-set categories by using DALL-E. During testing, they employ DINO and CLIP to get more accurate predictions by matching a test image to the generated images and, by matching a test image to category embeddings. They validate the proposed method on public benchmarks.

**Strengths:**

This paper has the following strengths.

S1: The problem tackled, open-set object recognition is interesting and challenging.

S2: The proposed approach is reasonable. Using generative models to leverage knowledge-base and image generation capability could improve open-set recognition performance by mitigating spurious correlations between the category and features.

S3: The proposed method shows favorable performance on public benchmarks, without any fine-tuning.

S4: The ablation experiments show the effectiveness of the proposed design choices. Chain-of-thought reasoning, self-checking, and CLIP-based feedback all contribute to the final performance. Furthermore, the performance improvement is not merely from the category overlap between the close-set and the virtual open-set as shown in Table 6. The improvement might come from mitigated spurious correlation.


**Weaknesses:**

This paper has the following weaknesses.

W1: Although the proposed method is effective in open-set recognition, the technical contribution is somewhat limited. This is okay if the performance improvement compared to the baseline is very significant. However, the performance improvement compared to the baseline is not surprising given that the proposed method consists of four foundation models: GPT, DALL-E, CLIP, and DINO. For example, the proposed method shows 3.2 points AUROC improvement and 5.5 points OSCR improvement over the softmax baseline in TinyImageNet.

W2: The authors claim the improved performance comes from mitigating the spurious correlation. However, there is no quantitative evaluation on the amount of spurious correlation mitigated. It would be better to show some quantitative evidence of how much the proposed method mitigates the spurious correlation.


**Questions:**

Please address the weaknesses part. I have a few additional questions below.

Q1: What is the performance if we learn a few parameters on top of the foundation models used? For example, what is the performance if we add some adapter layers on top of CLIP and/or DINO to fine-tune the model on the target datasets? Can we get an even stronger performance?

Q2: Does the proposed method work well on more fine-grained recognition tasks? E.g., CUB or Oxford Flowers datasets. It seems that the proposed method heavily relies on ChatGPT’s common sense reasoning ability. What happens if ChatGPT cannot generate reasonable descriptions of certain fine-grained classes? Since the proposed method does not learn anything, it might fail if ChatGPT fails. I would like to listen to the authors' opinions and/or empirical validation on this issue.


**Limitations:**

I could not find limitation and broader impact part.

---

> ### Author Rebuttal · Authors · 2023-08-10
>
> >*Q1: Technical contribution.*
>
> **A1:** In this work, we propose a novel framework that collaborates large models and performs training-free open-set object recognition. Moreover, to effectively extract implicit knowledge from large models, we also incorporate several novel designs, such as iterative self-checking and cyclic self-assessing. To the best of our knowledge, we are the first to collaborate different large models to perform open-set object recognition.
>
> >*Q2: The proposed method consisting of four foundation models only outperforms baseline by 3.2 and 5.5 points.*
>
> **A2:** We would like to clarify, similar to our method, the softmax baseline we set up also collaborates foundation models (as described in Line 320-324 in paper). This softmax baseline can be actually regarded as a variant of our method without designs to extract implicit knowledge, and the performance improvement (3.2 and 5.5) shows the benefit of the designs to extract implicit knowledge, instead of fully showing the superiority of our method. Note that the baseline (with large models) outperforms previous SOTA methods [6] by 9.2 on OSCR, and our designs further improve over baseline by 5.5 on OSCR, showing that overall, our collaboration of large models and designs bring large improvement (9.2 + 5.5 = 14.7). Sorry for confusion caused. We will make it clearer in paper.
>
> >*Q3: Show some quantitative evaluation on the amount of spurious correlation mitigated.*
>
>
> **A3:** While it can be difficult to directly and precisely evaluate spurious correlation, below, we still do some analysis and show some quantitative evaluation. Take the open-set class Crane as an example, without our method's list of virtual open-set classes used, 40% of Crane are misrecognized as closed-set class Albatross (both with white feathers) and 26% of Crane are misrecognized as closed-set class Black Stork (both with long lags). While after adding virtual open-set classes such as Swan (also with white feathers like Albatross) and Avocet (also with long legs like Black Stork), only 4% of Crane are misrecognized as Albatross and 2% of Crane are misrecognized as Black Stork. Such reduced amounts (40% to 4% and 26% to 2%) can imply the amounts of spurious correlation mitigated. Similar to what was done above, we measure the reduced amount for all classes on the TinyImageNet protocol. We find that, the amount reduced (mitigated) on average is around 24% after applying our method, implying the efficacy of our method in mitigating spurious correlation.
>
> >*Q4: Add adapter layers on top of CLIP and/or DINO.*
>
> **A4:** We add two adapter (linear) layers on top of both CLIP and DINO, and fine-tune these layers on the training set of the target dataset while keeping other parts frozen. We fine-tune CLIP (or not) and fine-tune DINO (or not) to get the four variants below. We report AUROC.
>
> | Methods  | CIFAR10 | CIFAR+10 | CIFAR+50 | TinyImageNet |
> |-|-|-|-|-|
> | **CLIP & DINO**  | 96.6 | 98.9 | 98.5 | 86.7|
> | **Fine-tuned CLIP & DINO**  | 96.3 | 98.5 | 98.0 | 86.8 |
> | **CLIP & fine-tuned DINO**  | 96.4 | 98.4 | 98.2 | 86.8 |
> | **Fine-tuned CLIP & fine-tuned DINO**  | 96.2 | 98.3 | 97.9 | 86.9 |
>
> As shown, on smaller-scale datasets (CIFAR10, CIFAR+10, and CIFAR+50), fine-tuning leads to slight performance drop, while on the relatively large TinyImageNet, fine-tuning leads to slight performance enhancement. Nevertheless, without any fine-tuning, our training-free method has already outperformed previous works largely.
>
> >*Q5: Fine-grained recognition tasks (CUB or Oxford Flowers datasets)*
>
> **A5:** Below, we evaluate on the more fine-grained datasets CUB and Oxford Flowers, and compare our method with previous SOTA methods. We report AUROC.
>
> | Methods  | CUB-Easy | CUB-Hard |
> |-|-|-|
> | **MLS [48]**  | 88.3 | 79.3 |
> | **Ours**  | **90.0** | **81.4** |
>
> | Methods  | Oxford Flowers |
> |-|-|
> | **DML [a]**  | 90.8 |
> | **Ours**  | **91.7** |
>
> As shown, on these more fine-grained datasets, our method can also outperform previous SOTA methods. This further shows the efficacy of our method.
>
> [a] The Importance of Metric Learning for Robotic Vision: Open Set Recognition and Active Learning. ICRA, 2019.
>
> >*Q6: What happens if ChatGPT cannot generate reasonable descriptions of certain fine-grained classes?*
>
>
> **A6:** Among ChatGPT's generated descriptions, we observe that, reasonable descriptions are consistently generated for all the evaluated classes (including fine-grained classes), while for certain (fine-grained) classes, in rare cases, unreasonable descriptions are also generated. But note that in our work, we designed a cyclic self-assessing module to replace unreasonable descriptions with reasonable descriptions. We find that, before passing into the self-assessing module, 3% of descriptions are checked to be unreasonable, but none of the descriptions output by the module is checked to be unreasonable. This shows that ChatGPT has small probability to generate unreasonable descriptions, while our self-assessing module can further mitigate this problem. The above check is done by inviting 3 volunteers and passing the same 1000 descriptions generated from fine-grained classes in CUB to each of them. The 3 volunteers first make decisions independently and then discuss disagreed decisions. We will also add above analysis to Supp.
>
> Though not observed, if ChatGPT cannot generate any reasonable description of a certain fine-grained class, all generated (unreasonable) descriptions may fail to be replaced in self-assessing process and images can fail to be generated for this class. This can weaken the mitigation of spurious-discriminative features w.r.t. this class. Nevertheless, in our experiments, ChatGPT is observed to be able to generate reasonable descriptions for all the evaluated classes, and our method achieves SOTA performances on all splits and all datasets, showing the efficacy of our method.

---

> > ### Comment · Reviewer_zWHw · 2023-08-16
> > **Acknowledging the rebuttal and other reviews**
> >
> > I have read the rebuttal and other reviews. The rebuttal from the authors resolved most of my concerns. I appreciate the clarification on the softmax baseline and the additional analysis on the spurious correlation. The proposed method seems to have merit. I am leaning toward accepting this paper. I am increasing my rating to 6.

---

> > > ### Author Response · Authors · 2023-08-21
> > >
> > > Respectful Reviewer zWHw,
> > >
> > > We are glad that we have resolved most of your concerns. Thanks for your time and effort and thank you for recommending accepting our paper.
> > >
> > > Best regards,
> > >
> > > Authors

---

### Official Review · Reviewer_BEEJ · 2023-07-05

**Soundness:** 3 good
**Presentation:** 3 good
**Contribution:** 3 good
**Rating:** 6
**Confidence:** 4

**Summary:**

This work demonstrates a very interesting and sophisticated usage of various large models, including ChatGPT, CLIP, DINO and DALL-E., to tackle the open-set recognition problem. The main challenge in open-set recognition is that the chosen classifier may get confused by certain spurious-discriminative features that are shared between the closed- and open-set classes. The main idea of this work is to leverage the text-to-image alignment capability of CLIP and image-to-image alignment capability of DINO with the aid of virtual open-set classes. The list of virtual open-set classes is elicited from LLM, i.e. ChatGPT, through carefully designed prompts that introduces intermediate reasoning and self-checking. Additionally, the descriptive text can then serve as prompts for DALL-E to synthesize the representative images of the corresponding open-set classes. During inference time, both the pre-trained CLIP and DINO models can then utilize the text class names and synthetic representative images to perform zero-shot (training-free) classification, and meanwhile reducing the influence of spurious-discriminative features. Evaluation on several benchmarks demonstrates new state-of-the-art performance.

**Strengths:**

1. The paper is well-written and very easy to follow. The additional information provided in the supplemental material is also very helpful in resolving most of the doubts I had when reading the main manuscript. In the beginning, I particularly felt that how the large models collaborate and self-improve to accomplish their jobs was a little magical and I wanted some more transparency after reading the main manuscript. I appreciate that the authors had put sufficient information in the supplemental material to make the whole work more technically sound and convincing.
2. This work tackles the open-set recognition from the system-level and demonstrates a very clever usage of several off-the-shelf large models of different modalities. Although it didn't introduce new techniques to improve any of the models involved, it successfully combined them to achieve superior performance. From the practical point of view, this method possesses many advantages, such as being training free and fast inference, that are important in real-world applications.

**Weaknesses:**

1. Although large model collaboration is an interesting way of solving open-set recognition problem, it treats the large models as black boxes as it is. and it is perhaps not easy to come up with an even more sophisticated method to build a LMC system like this work, In some sense, I feel that it does not pave a new way for follow-up research, and yet I do not believe the open-set recognition problem is completely solved by the LMC approach.
2. Although I don't think the evaluation of this work is insufficient, I think it is still worth validating the effectiveness of LMC framework on larger benchmarks such as the Semantic Shift Benchmark proposed in [48]. I doubt if the construction of virtual open-set classes by LLM will fail at a certain point as the number of fine-grained visually similar close-set classes grows. Another drawback of using LLM is that it is very difficult for human to "fact check" the answers and detect any content that is made up by LLMs.

**Questions:**

From the exemplar dialogs with ChatGPT, the conversation still seems a bit open-ended and unstructured. I am curious about how the authors completed the preparation of the virtual open-set list, synthesis of diverse representative images and pre-compute the CLIP and DINO features in just about 38.9 minutes (Appendix, Table 9). I assume that this is done programmatically. Are there any tricks involved here to simplify the post-processing of ChatGPT answers? As far as I know, the most important tunable parameter when calling OpenAI API is the temperature, which controls the diversity of the results that the LLM generates. How did you set this parameter? Does it affect the quality of ChatGPT's self-checking process?

**Limitations:**

The performance of LMC framework is apparently affected by the large models and there is no discussion about it. I am particularly curious about the potential failure modes of ChatGPT.

---

> ### Author Rebuttal · Authors · 2023-08-10
>
> >*Q1: Pave a new way for follow-up research.*
>
> **A1:** Thank you for pointing out our method *is an interesting way*. Below, we also discuss potential **follow-up research** in solving open-set recognition (OSR) that could be inspired by our work. In our method, we identify suitable large models, design schemes to extract implicit knowledge, and design collaboration of large models to solve OSR problem. This implies that follow-up research work can further investigate (1) how to identify more suitable large models, (2) how to better extract knowledge from large models, and (3) how to better collaborate large models for OSR. Besides, how to collaborate large models with conventional OSR methods to further improve performance with the complimentary ability can also be interesting. Moreover, despite the convenience of training-free, adding trainable modules to further improve the performance can be explored. This implies that our method could also inspire new ways for follow-up research. We will add more detailed discussions to paper.
>
> >*Q2: (1) Validate on larger benchmark Semantic Shift Benchmark. (2) Construction will fail at a certain point as the number of classes grows. (3) Difficult for human to "fact check".*
>
> **A2:** (1) **Larger benchmarks.** We evaluate on a larger benchmark Semantic Shift Benchmark [48] and report AUROC.
>
> |Methods|CUB-Easy|CUB-Hard|SCars-Easy|SCars-Hard|FGVC-Aircraft-Easy|FGVC-Aircraft-Hard|ImageNet-Easy|ImageNet-Hard|
> |-|-|-|-|-|-|-|-|-|
> |**MLS [48] (ICLR 2022)**|88.3|79.3|94.0|82.2|90.7|82.3|78.7|72.8|
> |**Ours**|**90.0**|**81.4**|**96.1**|**84.3**|**92.1**|**84.1**|**80.0**|**74.8**|
>
> As shown, on larger benchmark that has many fine-grained classes, our method can also outperform the previous SOTA method on all splits.
>
> (2) **Construction of classes.** We admit that if there are huge amount of very similar fine-grained classes, construction of classes may yield weaker performance. However, even on large fine-grained datasets (Semantic Shift) with 1000 close-set classes, we observe that our method can still construct classes reliably and achieve SOTA performance, showing the robustness of our method. But we also need to mention that, if there are too many and too similar fine-grained classes, it will become very difficult for existing open-set recognition methods in common.
>
> (3) **Fact check.** To further show the efficacy of our designs, below, we also introduce **human fact check**. We invite 3 volunteers, each conducting fact checks of the construction of classes independently with access to internet and online bases (e.g., Wiki) to help their assessment. Then, volunteers further discuss the disagreed class names. We conduct above fact check on the constructed classes for CIFAR10. We observe that, without proposed designs (intermediate reasoning and self-checking), around 5.5% constructed classes are judged as low quality, while with proposed designs, only around 0.9% constructed classes are judged as low quality. This further shows (1) the efficacy of proposed designs, (2) though it is difficult to do very accurate fact check, we can still perform fact check to some extent.
>
>
> >*Q3: ChatGPT's conversation seems unstructured. How to complete preparation in just about 38.9 mins. Any tricks to simplify the post-processing of ChatGPT answers?*
>
> **A3:** As mentioned in Supp, we complete the preparation automatically with a single script. Using this script (Python), we can (1) structure ChatGPT's answers and (2) complete the preparation very efficiently and by the following processes:
>
> (1) Like online tutorial [a], we simply use ChatGPT to **structure the answers** by instructing ChatGPT: "Please respond in a list and start each class with '--'". Thus, the answers are structured.
>
> (2) As the script can run automatically, we can complete preparation conveniently. To further enhance the efficiency, in our script, we open 20 ChatGPT instances concurrently during simulating class names, and use 20 DALL-E instances concurrently to synthesize images. Using this way, on a consumable workstation with a single 3090 GPU, we can **finish the preparation in just about 38.9 mins**.
>
> We will release code and script.
>
> [a] Zdnet. 7 advanced ChatGPT prompt-writing tips you need to know.
>
>
> >*Q4: How did you set the temperature? Does it affect self-checking?*
>
> **A4:** We set the temperature to its default value 1.0. Below, we test setting it to different values in self-checking.
>
> |Temperature|0.1|0.4|0.7|1.0|1.3|1.6|1.9|
> |-|-|-|-|-|-|-|-|
> |AUROC|86.1|86.4|86.6|86.7|86.7|86.3|86.0|
>
> We observe that our method gets optimal results on 1.0 and 1.3 (1.0 used in paper). We also find that setting temperature to different values all outperform the variant w/o self-checking, showing the efficacy of self-checking.
>
>
> >*Q5: The performance is affected by the large models and there is no discussion about it. Curious about potential failure modes of ChatGPT.*
>
> **A5:** (1) **The performance is affected by the capability of each large model**. For example, GPT has different versions (e.g., GPT 3.0 and 3.5) with different capabilities. For GPT 3.0, AUROC on TinyImageNet is 85.2, while for GPT 3.5, AUROC is 86.7. This shows that the performance is affected by the large models. We will add more discussion to paper.
>
> (2) **Potential failure mode** of ChatGPT: We observe that when we ask ChatGPT a complex question (e.g., "Given a list of classes [classes], please list some classes that share spurious-discriminative features with class [class]?"), on some classes, ChatGPT fails to well understand the question and provides undesired answers: Sorry, as an AI language model, I cannot answer this question. This can be seen as a failure mode. Nevertheless, as discussed in Sec. 3 in paper, we design to guide ChatGPT step-by-step with intermediate reasoning. When adding this design, we well-handle this failure mode and observe that ChatGPT can stably generate good-quality answers.

---

> > ### Comment · Reviewer_BEEJ · 2023-08-11
> > **Response to authors' rebuttal**
> >
> > I appreciate the great efforts made by the authors to prepare the rebuttal. I was impressed by the superior performance shown in Semantic Shift Benchmark and other experiments in the responses to other reviews. The rebuttal addressed most of my concerns and questions. I still hold my opinion that the LMC framework leaves less room for future exploration because large models remain a black box and attempts in (1), (3) of A1 could be easily regarded as *tricks* or lack of technical contribution. Nevertheless, I am in favor of accepting this paper and will keep my rating as Weak Accept.

---

> > > ### Author Response · Authors · 2023-08-21
> > >
> > > Respectful Reviewer BEEJ,
> > >
> > > Thank you for recommending accepting our paper. Besides the above-mentioned follow-up research in A1, one more thing that can be explored in the future is to incorporate black-box optimization algorithms (e.g., CMA Evolution Strategy) into our LMC framework. Recall that the LMC framework we propose in our paper is a training-free framework. Despite the convenience of being training-free, the usage of black-box large models in our framework for open-set recognition can be further improved through black-box optimization. Some attempts to enhance the usage of a large model for a specific task through black-box optimization have been made in the NLP area (e.g., "Black-Box Tuning for Language-Model-as-a-Service" published on ICML 2022). In our framework, for example, one potential way to enhance the usage of large models can be to utilize black-box optimization to craft prompts for ChatGPT in the virtual open-set class simulation process. By doing so, the implicit knowledge of ChatGPT can be better extracted. We will discuss this as a future direction in our paper as well.
> > >
> > > Thank you again for your time and effort.
> > >
> > > Best regards,
> > >
> > > Authors

---

### Official Review · Reviewer_Qabu · 2023-07-05

**Soundness:** 3 good
**Presentation:** 3 good
**Contribution:** 3 good
**Rating:** 6
**Confidence:** 4

**Summary:**

This paper proposed to incorporate several large models to solve the spurious-discriminative problem in open-set recognition. It first prompts ChatGPT to describe the known classes and generate a list of new classes that share the spurious-discriminative features with the known classes. Then it prompts DALL-E to generate the images of known and unknown classes. It also uses CLIP to detect the less accurate generated images and asks ChatGPT to refine the description and uses DALL-E to generate again. During inference, it uses both the CLIP model and DINO model to calculate the uncertainty scores and fuse their results. The experiments and ablation study are comprehensive. Overall, I think this paper provides a new alternative solution for open-set recognition, and the data generation process could be very useful for the industry.

**Strengths:**

1. The overall idea of using several large models for training-free open-set recognition is new and interesting. The pipeline of using ChatGPT to generate the new classes that share the spurious-discriminative features with the known classes is reasonable. The refinement procedure using CLIP and ChatGPT is very rigorous and effective.
2. The inference pipeline that fuses the results of CLIP and DINO is also reasonable. It is somewhat similar to the method in the few-shot open-set recognition [1, 2], which also fuses the uncertainty score from two aspects. The authors may consider citing the related papers.
3. The experiments and ablation studies are very comprehensive.

Reference:
[1] Ssd: A unified framework for self-supervised
outlier detection. In ICLR, 2022.
[2] THE DEVIL IS IN THE WRONGLY-CLASSIFIED SAMPLES: TOWARDS UNIFIED OPEN-SET RECOGNITION. In ICLR 2023.

**Weaknesses:**

The main pipeline is a kind of data generation process, which could not be done online. So although it is a good try to use several large models and coordinate them to generate wanted data, it is more like an engineering project rather than a research topic.

**Questions:**

1. How much time does the data generation cost for different datasets?
2. What if we use other pre-trained image models instead of DINO? Like ImageNet pre-trained or MoCo?

**Limitations:**

None.

---

> ### Author Rebuttal · Authors · 2023-08-10
>
> >*Q1: May consider citing the related papers [1,2].*
>
> **A1:** We will cite [1,2] in the paper. Note that different from [1,2] which fuse the uncertainty score from two aspects (e.g., from FS-KNN and Softmax), we are the first to collaborate large models and leverage their rich and distinct knowledge to perform open-set object recognition. During inference, as mentioned in our paper, we collaborate two large models (CLIP and DINO) complementarily to match the testing image to both the overall concept and detailed local features of each class.
>
> >*Q2: The main pipeline is a kind of data generation process, which could not be done online. [...] it is more like an engineering project rather than a research topic.*
>
> **A2:** (1) **Online**: In open-set recognition, most existing works rely on a training process. Besides training, many works [8,13,21,30,32] also perform data generation (i.e., they need both training and data generation). The data generation process in these works (and also in our method) is not done **online**. In contrast to previous works, our framework performs **data generation** but does not require any training process. In other words, in this work, we, for the first time, propose a **training-free** framework to handle the open-set recognition task. This shows the advantage of our research work. As also mentioned by Reviewer BEEJ, "this method possesses many advantages, such as being training-free and fast inference, that are important in real-world applications".
>
> (2) **Research topic**: Open-set object recognition is a challenging research problem and has attracted lots of research attention. As mentioned by Reviewer zWHw, "open-set object recognition is interesting and challenging".
>
> To perform open-set object recognition well, as also pointed out by Reviewer BEEJ, "the main challenge in open-set recognition is that the chosen classifier may get confused by certain spurious-discriminative features". To tackle this challenge, in our paper, inspired by that different large models often contain rich and distinct knowledge, we, for the first time, propose a framework to collaborate different large models to handle the open-set object recognition task. Reviewer BEEJ also mentions that, our work "demonstrates a very clever usage of several off-the-shelf large models of different modalities".
>
> However, utilizing and collaborating large models to perform open-set object recognition is challenging and is a research problem by itself that is worth exploring. To handle this problem and extract implicit knowledge in large models effectively, in our paper, we also incorporate our framework with several novel designs. During simulating names of virtual open-set classes, to simulate a more comprehensive list of virtual open-set classes to better cover the spurious-discriminative features, we propose to enable the large model to perform self-checking. Moreover, during generating images via DALL-E, to ensure that the images generated can well represent their desired classes, we also design an effective cyclic self-assessing module to perform cross-checking across different large models and refine the generated images iteratively.
>
> In summary, open-set object recognition is challenging, and one of its main research challenges is to handle spurious-discriminative features (as mentioned by Reviewer BEEJ). In this work, to handle this research challenge, we, for the first time, propose a framework to collaborate large models to perform open-set object recognition. We also incorporate our framework with several new and effective designs to utilize and collaborate large models better. Our framework achieves SOTA performance on all the evaluated benchmarks. **These show that, our work can bring some new research contributions**.
>
> >*Q3: Data generation time cost.*
>
> **A3:** In our framework, data generation consists of two steps: (1) generating the names of virtual open-set classes and (2) generating diverse images w.r.t. each class. Below, we report the time cost on each of the above two steps across different datasets.
>
> | Methods | CIFAR+10 | CIFAR+50 | CIFAR10 | TinyImageNet |
> |---|---|---|---|---|
> | **Name generation**  | around 1.3 mins | around 1.3 mins | around 2.0 mins | around 5.2 mins |
> | **Image generation**  | around 9.4 mins | around 9.4 mins | around 9.5 mins | around 32.7 mins |
>
> As shown, our framework does not need much time during both the above two steps. Taking TinyImageNet as an example, on our workstation with a single RTX 3090 GPU, the above two steps take 37.9 minutes in total. Note that after the above two steps, our framework is free of training, while the training stage of previous SOTA methods often requires a long training time (e.g., ZOC [12] takes around 40 hours during its training stage when run on our workstation).
>
> >*Q4: Other pre-trained image models instead of DINO like ImageNet pre-trained or MoCo.*
>
> **A4:** Below, we compare "our framework with DINO" with two other variants. In these two variants, we replace DINO with ImageNet pre-trained transformer and MoCo respectively, while keeping the rest parts of our framework to be the same. We report the AUROC score.
>
> | Methods  | CIFAR10 | CIFAR+10 | CIFAR+50 | TinyImageNet |
> |---|---|---|---|---|
> | **Previous SOTA** | 95.1 | 97.9 | 97.6 | 84.6 |
> | **ImageNet pre-trained transformer**  | 95.7 | 98.5 | 98.1 | 86.4 |
> | **MoCo**  | 95.2 | **98.9** | 98.0 | 85.8 |
> | **DINO**  | **96.6** | **98.9** | **98.5** | **86.7** |
>
> As shown, no matter using DINO, ImageNet pre-trained transformer, or MoCo, our framework can consistently outperform previous SOTAs across different datasets. Moreover, our framework with DINO outperforms the other two variants. This can be because, as discussed in our paper, the collaboration of DINO with good capability in detailed local features, and CLIP with good capability in overall concepts, is complementary and can achieve good performance.

---

> > ### Comment · Reviewer_Qabu · 2023-08-11
> > **Response to the rebuttal**
> >
> > I carefully read the rebuttal and the reviews of other reviewers, and I believe that my concerns are solved. This work is valuable for the research community and industry, so I raise my score to 6.

---

> > > ### Author Response · Authors · 2023-08-21
> > >
> > > Respectful Reviewer Qabu,
> > >
> > > Thank you for your time and effort and thank you again for pointing out that this work is valuable for both the research community and industry.
> > >
> > > Best regards,
> > >
> > > Authors

---

### Official Review · Reviewer_Kpbj · 2023-07-06

**Soundness:** 2 fair
**Presentation:** 3 good
**Contribution:** 3 good
**Rating:** 5
**Confidence:** 5

**Summary:**

This paper proposes an open-set recognition framework, Large Model Collaboration (LMC), which collaborates several large models (ChatGPT, DALL-E, CLIP, and DINO) to make use of the rich implicit knowledge to reduce the reliance on spurious-discriminative features. The proposed framework consists of the following two stages:

(i)	The first stage is to simulate virtual open-set classes, including simulating names and generating images. In order to improve the effectiveness of the simulated names for open-set classes, ChatGPT is asked with three designed questions and the corresponding intermediate rationales for obtaining names of new classes that share spurious-discriminative features with each closed-set class, and is guided to perform iterative self-check (where the new simulated class names are used to expand the name list, and the expanded name list is used to obtain new simulated class names) for covering as many spurious-discriminative features as possible. Then, ChatGPT is asked to generate diverse descriptions for each class in the expanded name list, and DALL-E is used to generate images based on these descriptions. In order to improve the accuracy of the generated images, a cyclic self-assessing module is proposed, where ChatGPT is provided with feedbacks from CLIP about the less accurate images to refine the descriptions generated by ChatGPT and DALL-E is used to re-generate images based on the refined descriptions. At this stage, an expanded list of both closed-set and simulated open-set classes and diverse generated images for each class in the list are obtained.

(ii)	The second stage is for inference based on the expanded list and the generated images. At this stage, two alignments (image-text alignment by CLIP and image-image alignment by DINO) are performed for obtaining the corresponding scores to each testing image, and the weighted result of the two scores is used for open-set recognition.



**Strengths:**

(1)	The idea, which makes use of several large models for open-set recognition, is somewhat novel and interesting.

(2)	The writing is clear and easy to follow. Some examples are well visualized.

(3)	Experimental results on four small-scale datasets (CIFAR10, CIFAR+10, CIFAR+50, and TinyImageNet) demonstrate that the proposed methods performs better than the comparative methods.


**Weaknesses:**

(1)	Lacking comparisons under the cross-dataset setup: The proposed framework is only evaluated under the standard setup where the open-set-class images and closed-set-class images are from a same dataset. Such experiments are insufficient, and in fact, many open-set recognition methods [15,21,36,45,55,63] have also been evaluated under the cross-dataset setup.

(2)	Lacking comparisons on larger-scale datasets: in this paper, only four small-scale datasets (i.e., CIFAR10, CIFAR+10, CIFAR+50, and TinyImageNet) are used for evaluation. It is necessary to make comparisons on some larger-scale datasets, e.g., CUB and FGVC-Aircraft.

(3)	Comparison with other image generative methods: One main contribution of the proposed framework is generating virtual open-set-class images. Many other works (e.g., [8,21,30]) also generate virtual images, and it is better to replace the image generation method used in the proposed framework with some of them, and make a further comparison.

(4)	Lacking details of efficiency: One advantage of the training-free methods is its high efficiency. In the proposed framework, although there is no training stage, the pre-processing stage and the inference stage which is based on two alignments may also bring additional computational costs. Hence, it is suggested to provide details about the efficiency of the two stages.

(5)	Lacking discussion about the ablation results: In Sec 4.4, several ablation studies are conducted. However, the authors only stated that the final results were improved, but didn’t give any discussion on the effect of each module.


**Questions:**

Please see the weaknesses above.

**Limitations:**

The authors did not adequately address the limitations.
Two additional suggestions are listed here:
(1) It’s better to add the full name of LMC, i.e., Large Model Collaboration, in the abstract in line 6.

(2) In Line 219, it is suggested to add a brief description of 5, although Fig. 5 is also described in Line 415.

---

> ### Author Rebuttal · Authors · 2023-08-10
>
> >*Q1: Comparisons under the cross-dataset setup.*
>
> **A1:** Below, we evaluate our method under a common cross-dataset setup following [15,36,45,55,63]. In this setup, closed-set-class images are from CIFAR-10, and open-set-class images are from ImageNet-crop, ImageNet-resize, LSUN-crop, and LSUN-resize. We use the same evaluation metric F1-score as [15,36,45,55,63] under this setup.
>
> | Methods  | ImageNet-crop | ImageNet-resize | LSUN-crop | LSUN-resize |
> |---|---|---|---|---|
> | **CVAECapOSR [15]**| 85.7 | 83.4 | 86.8 | 88.2|
> | **GFROSR [36]**  | 82.1 | 79.2 | 84.3 | 80.5 |
> | **CGDL [45]**  | 84.0 | 83.2 | 80.6 | 81.2 |
> | **CROSR [55]**  | 73.3 | 76.3 | 72.0 | 74.9 |
> | **PROSER [63]**  | 84.9 | 82.4 | 86.7 | 85.6 |
> | **Ours**  | **88.0** | **86.0** | **91.5** | **93.5** |
>
> As shown, under the cross-dataset setup, our method also achieves significant performance improvement over previous methods, further demonstrating the effectiveness of our method.
>
> >*Q2: Comparisons on larger-scale datasets, e.g., CUB and FGVC-Aircraft.*
>
> **A2:** Following your suggestion, below, we also evaluate our method on larger-scale datasets including CUB and FGVC-Aircraft, and compare our method with previous methods which report results on these datasets. We report the AUROC score and we follow the Easy/Hard data split of [48] for these two datasets.
>
> | Methods  | CUB-Easy | CUB-Hard | FGVC-Aircraft-Easy | FGVC-Aircraft-Hard |
> |---|---|---|---|---|
> | **ARPL+CS [6] (TPAMI 2021)**  | 83.5 | 75.5 | 87.0 | 77.7 |
> | **MLS [48] (ICLR 2022)**  | 88.3 | 79.3 | 90.7 | 82.3 |
> | **Ours**  | **90.0** | **81.4** | **92.1** | **84.1** |
>
> As shown, on these larger-scale datasets, our method can also outperform previous methods and achieve SOTA performance on all splits.
>
> >*Q3: Replace the image generation method: Many other works (e.g., [8,21,30]) also generate virtual images.*
>
> **A3:** Following your suggestion, below, we compare our framework with three variants of our framework (**Our other parts + Image generation with [8]**, **Our other parts + Image generation with [21]**, **Our other parts + Image generation with [30]**).  In these three variants, we replace the image generation method used in our framework with the image generation method in [8], [21], and [30] respectively, while keeping the rest parts of our framework to be the same (e.g., in these three variants, we will still use ChatGPT to simulate a list of virtual open-set classes, and these classes will still be aligned with the testing image through CLIP during inference).
>
> | Methods  | CIFAR10 | CIFAR+10 | CIFAR+50 | TinyImageNet |
> |---|---|---|---|---|
> | **Our other parts + Image generation with [8]**  | 95.0 | 97.1 | 96.1 | 84.5 |
> | **Our other parts + Image generation with [21]**  | 94.3 | 96.7 | 95.3 | 83.7 |
> | **Our other parts + Image generation with [30]**  | 94.5 | 97.0 | 95.4 | 83.8 |
> | **Ours**  | **96.6** | **98.9** | **98.5** | **86.7** |
>
> As shown, our framework with its original image generation method outperforms all three variants, showing the effectiveness of the image generation method proposed in our framework.
>
> >*Q4: Details of efficiency: it is suggested to provide details about the efficiency of the two stages.*
>
> **A4:** Below, we show the time of the two stages, i.e., (1) the pre-processing stage and (2) the inference stage. We evaluate the time cost of our framework on a consumable workstation with an RTX 3090 GPU. With respect to the pre-processing stage, the time cost of our framework ranges from around 10.9 minutes to 38.9 minutes across the evaluated datasets. Taking TinyImageNet as an example, on our workstation, the pre-processing stage of our framework takes 38.9 minutes. Note that our framework is free of training, while the training stage of previous SOTA methods often requires a long training time (e.g., ZOC [12] takes around 40 hours during its training stage when run on our workstation). This demonstrates the efficiency of our training-free framework. Moreover, with respect to the inference stage, we find that our method achieves real-time performance (around 0.02 seconds per image across the evaluated datasets).
>
> >*Q5: Discussion about the ablation results in Sec 4.4 on the effect of each module.*
>
> **A5:** Following your suggestion, we will add discussions to Sec 4.4 on the effect of each module. Below, we take Tab. 4 and Tab. 5 in Sec 4.4 as examples.
>
> (1) In Tab. 4, our framework that has a design to guide ChatGPT with intermediate reasoning outperforms the variant "w/o intermediate reasoning". This is because, the intermediate reasoning design can lead ChatGPT to better understand the asked question, and thus lead to performance improvement. Moreover, our framework with the self-checking design also outperforms the variant "w/o self-checking". The performance improvement is due to the effect of the self-checking module, which can lead ChatGPT to simulate a more comprehensive list of virtual open-set classes.
>
> (2) In Tab. 5, our framework that uses the cyclic self-assessing module outperforms all three variants, i.e., the "w/o self-assessing" variant, the "Check and discard" variant, and the "Check and naively refine" variant. Such improvement can be due to the effect of the cyclic self-assessing module, which can leverage CLIP to provide specific feedback for ChatGPT, so that ChatGPT can refine the descriptions more toward the direction of our desire. Note that all three variants generate images in some alternative way but do not use the cyclic self-assessing module.
>
> Due to space limitations here, we will add more detailed discussions w.r.t. all ablations to Sec 4.4 of paper.
>
>
> >*Q6: Two additional suggestions.*
>
> **A6:** Thanks for your suggestion. We will (1) add the full name of LMC, i.e., Large Model Collaboration, to the abstract, and (2) describe Fig. 5 in Line 219 in the revised version.

---

> > ### Comment · Reviewer_Kpbj · 2023-08-19
> >
> > I appreciate the authors’ responses that have addressed most of my concerns. To be honest, it is better to make a cross-dataset comparison with some methods (e.g. [27, 12]) that have appeared and performed relatively better in the original tables 1 and 2 of the submitted text, rather than some newly added methods. Just as stated in my first-round comment, I still feel that the idea of this paper is somewhat interesting, so I will keep my rating as Borderline accept.

---

> > > ### Author Response · Authors · 2023-08-21
> > >
> > > Respectful Reviewer Kpbj,
> > >
> > > We are glad that we have addressed most of your concerns and thank you for leaning towards accepting this paper.
> > >
> > > With respect to the cross-dataset comparison, as pointed out by you in the first-round comment, "many open-set recognition methods [15,21,36,45,55,63] have also been evaluated under the cross-dataset setup." Among these methods, a common cross-dataset setup is utilized by most of them ([15,36,45,55,63]), in which closed-set-class images are from CIFAR-10, and open-set-class images are from ImageNet-crop, ImageNet-resize, LSUN-crop, and LSUN-resize. Following [15,36,45,55,63], we also evaluate our method under this common cross-dataset setup, and we compare our method with [15,36,45,55,63] during our rebuttal.
> > >
> > > Thanks for your suggestion that it is better to also make a cross-dataset comparison with methods in [12] and [27]. While [12] and [27] do not make cross-dataset comparisons in their own paper, below, we re-evaluate [12] and [27] under the above common cross-dataset setup and compare our method with them.
> > >
> > > | Methods  | ImageNet-crop | ImageNet-resize | LSUN-crop | LSUN-resize |
> > > |---|---|---|---|---|
> > > | **ZOC [12]**  | 84.6 | 81.8 | 87.4 | 86.8 |
> > > | **PMAL [27]**  | 85.8 | 83.2 | 86.5 | 87.6 |
> > > | **Ours**  | **88.0** | **86.0** | **91.5** | **93.5** |
> > >
> > > As shown, under the cross-dataset setup, our method also outperforms [12] and [27] by a large margin. This further shows the effectiveness of our method. We will add the above discussion and results to our paper as well.
> > >
> > > Once again, we would like to express our sincere thanks for your time and effort.
> > >
> > > Best regards,
> > >
> > > Authors

---

### Author Rebuttal · Authors · 2023-08-10

We thank all reviewers for recognition of our contributions (Reviewer Kpbj:"novel and interesting"; Reviewer Qabu:"the overall idea is new and interesting"; Reviewer BEEJ:"demonstrates a very clever usage of several off-the-shelf large models of different modalities", "possesses many advantages, such as being training free and fast inference, that are important in real-world applications"; Reviewer zWHw: "the problem tackled is interesting and challenging", "favorable performance").

---

### Decision · Program_Chairs · 2023-09-21

**Decision:**

Accept (poster)

**Comment:**

Four knowledgeable reviewers recommend acceptance of this paper citing it as a novel and valuable contribution for the research community. Two reviewers raised their rating after rebuttal and discussion. Authors should attend to main points in the reviews when preparing a final version. No basis to overturn reviews.